# Long-term sub-micron aerosol chemical composition in the boreal forest: inter- and intra-annual variability

Liine Heikkinen[1], Mikko Äijälä[1], Matthieu Riva[1,2], Krista Luoma[1], Kaspar Dällenbach[1], Juho Aalto[3], Pasi Aalto[1], Diego Aliaga[1], Minna Aurela[4], Helmi Keskinen[1], Ulla Makkonen[4], Pekka Rantala[1], Markku Kulmala[1], Tuukka Petäjä[1], Douglas Worsnop[1,5], and Mikael Ehn[1]

[1]Institute for Atmospheric and Earth System Research /Physics, Faculty of Science, University of Helsinki, Helsinki, FI–00014, Finland
[2]Univ Lyon, Université Claude Bernard Lyon 1, CNRS, IRCELYON, 69626, Villeurbanne, France
[3]Institute for Atmospheric and Earth System Research /Forest Sciences, Faculty of Agriculture and Forestry, University of Helsinki, Helsinki, FI–00014, Finland
[4]Atmospheric Composition Research, Finnish Meteorological Institute, Helsinki, FI–00101, Finland
[5]Aerodyne Research Inc., Billerica, MA, USA

Correspondence to: Liine Heikkinen (liine.heikkinen@helsinki.fi) and Mikael Ehn (mikael.ehn@helsinki.fi)

**Abstract.**

The Station for Measuring Ecosystem Atmosphere Relations (SMEAR) II is well known among atmospheric scientists due to the immense amount of observational data it provides of the earth–atmosphere interface. Moreover, SMEAR II plays an important role in large European research infrastructures, enabling the large scientific community to tackle climate and air pollution related questions, utilising the high-quality long-term data sets recorded at the site. So far, the well-documented site was missing the description of the seasonal variation of aerosol chemical composition that is crucial for understanding the complex biogeochemical and -physical processes governing the forest ecosystem. Here, we report the sub-micron aerosol chemical composition and its variability utilising data measured between 2012 and 2018 using an Aerosol Chemical Speciation Monitor (ACSM). We observed a bimodal seasonal trend in the sub–micron aerosol concentration culminating in February (2.7, 1.6, 5.1 µg m$^{-3}$ for median, 25$^{th}$, 75$^{th}$ percentiles, respectively) and July (4.2, 2.2, and 5.7 µg m$^{-3}$ for median, 25$^{th}$, 75$^{th}$ percentiles, respectively). The wintertime maximum was linked to an enhanced presence of inorganic aerosol species (ca. 50%) whereas the summertime maximum (ca. 80% organics) to biogenic secondary organic aerosol (SOA) formation. During the exceptionally hot Julys of 2014 and 2018, the organic aerosol concentrations were up to 70% higher than the 7–year July mean. The projected increase of heat wave frequency over Finland will most likely influence the loading and chemical composition of aerosol particles in the future. Our findings suggest strong influence of meteorological conditions such as radiation, ambient temperature, wind speed and direction on aerosol chemical composition. To our understanding, this is the longest time series reported describing the aerosol chemical composition measured online in the boreal region, but the continuous monitoring will be maintained also in the future.

## 1 Introduction

Both climate change and air pollution represent global grand challenges. Detailed monitoring of environments showing vulnerability towards them is crucial. The arctic and boreal forest are examples of such regions (Prăvălie, 2018;Kulmala, 2018). The boreal forest represents ~15% of the Earth's terrestrial area, spanning between 45 and 70 °N, and making up ~ 30% of the world's forests (Prăvălie, 2018). Over the course of the predicted warming, the boreal forest is likely to

move further north, resulting in arctic greening. In addition, the presence of southerly tree species are projected to increase
in the southern regions of the biome (Settele et al., 2014). These large–scale changes are linked to numerous complex
biogeochemical and -physical processes. These complexities greatly hamper our ability to make detailed predictions of
future changes, as exemplified by diversities in global model outputs of many important ecosystem- and climate-relevant
parameters (Fanourgakis et al., 2019). To help improve and constrain modelling efforts, comprehensive long-term high-
quality observational data are of utmost importance (Kulmala, 2018). Among the important parameters to monitor,
atmospheric composition, including both gaseous and particulate matter, provides a crucial link between the ecosystems
and climate.

Atmospheric aerosol particles affect Earth's radiative balance, influence ecosystems and human health, and reduce
visibility (Ramanathan et al., 2001;Boucher et al., 2013;Myhre et al., 2013). These particles can be emitted directly into
the atmosphere or form through gas-to-particle transition reactions from atmospheric vapours (Kulmala et al., 2004). The
composition of atmospheric aerosol particles has an extensive degree of variability depending on their origin. Their
composition covers a wide range of organic and inorganic species with differing physicochemical properties. These
properties affect the aerosol-related disturbances on the Earth's radiative forcing as salt particles scatter radiation
efficiently, whereas soot particles absorb it. In addition to this direct radiative effect, aerosol particles also participate in
cloud formation and processing. Indeed, every cloud droplet forms from an aerosol particle seed, termed a cloud
condensation nucleus (CCN). Moreover, these cloud seeds are often hygroscopic, which is directly linked to their
chemical composition. Important contributors to discrepancies estimating aerosol sensitivity of the boreal climate are
challenges in reproducing observations of aerosol chemical composition and properties (Fanourgakis et al., 2019).

Solar radiation and ambient temperature control both biogenic and human (anthropogenic) behaviour. In the northern
latitudes, the amount of radiation varies considerably in the course of a year yielding a large seasonal variation in ambient
temperature. The ambient temperature also fluctuates notably in diurnal scale. Ambient temperature influences emissions
of various biogenic volatile organic compounds (BVOCs) including monoterpenes (Guenther et al., 1993), whereas solar
radiation enables the photochemical reactions leading to oxidation products having lower volatilities. Secondary organic
aerosol (SOA) is formed from the partitioning of oxidised VOCs to the condensed phase. SOA is a key component in
tropospheric PM worldwide (Zhang et al., 2007a;Jimenez et al., 2009). While warmer conditions promote the emissions
of BVOCs, cold temperatures enhance the need of residential heating leading to emissions of primary particles as well as
a large variety of anthropogenic trace gases.

The major inorganic sub-micron aerosol species, of which the majority also originate from anthropogenic activities, are
sulphate, nitrate and ammonium (Zhang et al., 2007b;Jimenez et al., 2009). The presence of sulphur dioxide ($SO_2$), emitted
to the atmosphere from industrial processes and volcanic activity, has significantly increased compared to pre-industrial
conditions (Tsigaridis et al., 2006). It forms sulphuric acid upon oxidation that besides participating in the formation of
new particles, also readily condenses onto pre–existing aerosol particles increasing the particle mass loading and
ultimately modifying the acidity of atmospheric particles. Ammonia ($NH_3$), emitted in large quantities from industry and
agriculture, can partly neutralise particulate sulphuric acid forming ammonium sulphate, acknowledged as one of the
main contributors to sub–micron aerosol mass. In addition, ammonium nitrate, formed from the reaction between
ammonia and nitric acid is a common inorganic PM constituent. Nitrogen oxides ($NO_x = NO + NO_2$) from traffic
emissions and industry are the major nitric acid precursors in the atmosphere. These radicals play an important role in
atmospheric chemistry due to their high reactivity.

The concentrations of primary aerosol particles and the aerosol precursors (such as $NO_x$, $NH_3$, $SO_2$, and (B)VOCs) vary
during the year especially in the northern latitudes, where temperature differences between summer and winter are drastic.
Besides differences in the emissions, also the dynamics (thickness) of the atmospheric boundary layer influences airborne
pollutant concentrations. For example, during sunny summer days the boundary layer height can exceed two kilometres
height in the Northern hemisphere (McGrath-Spangler and Denning, 2013), and emissions from the surface are widely
dispersed in a large volume. During wintertime, the boundary layer can be more than a kilometre shallower than in
summer, and the pollutant loadings become concentrated closer to the surface. The boundary layer thickness is determined
by atmospheric stability. In unstable conditions, the air is rising and well mixed due to heating from below. In stable
conditions, generally caused by cooling from below, turbulence is supressed and mixing occurs only close to the Earth's
surface. Shallow, nocturnal boundary layers are often stable due to radiative cooling from the Earth's surface.

To conclude, the aerosol chemical composition and loading in the lower troposphere are highly dependent on different
emission sources and meteorological conditions. As these vary over the course of a year, also seasonal variation can be
expected in aerosol composition and loading. Importantly, variation also occurs invariably in inter–annual scale. For
example, year–to–year variation in ambient temperature is normal, but expected to increase with increased frequency of
climate extremes introduced by climate change (Pachauri and Meyer, 2014;Kim et al., 2018). Such variations could affect
air pollutant loadings in the boreal region, as milder winters might lead to a decrease of emissions from domestic wood
burning, and warmer summers might enhance the emissions from frequent and intense wild fires as well as promote SOA
formation from oxidised BVOCs. Inter–annual variability in aerosol composition and loading can also be introduced by
emission regulations. For example, atmospheric $PM_{10}$, $PM_{2.5}$, $SO_2$ and $NO_x$ concentrations have shown decreasing trends
in the past decades in United States and Europe (Wang et al., 2012;Aas et al., 2019;Anttila and Tuovinen, 2010;Simon et
al., 2014). Hence, for a well representative overview of aerosol climatology, and to truly capture seasonal variations
regarding air pollutants in the ruling boreal climate, a long–enough time series is required for analysis. Only then, a good
overview can be given of trends, variability, and seasonal and diurnal cycles of aerosol concentrations and composition.

The current study focuses on the seasonal variation of aerosol chemical composition and its year-to-year fluctuation at
the Station for Measuring Ecosystem – Atmosphere relations (SMEAR) II (Hari and Kulmala, 2005), located in the boreal
forest of Finland. The measurement period spans over seven years. SMEAR II is well known for its comprehensive,
simultaneous measurements tracking >1000 different environmental parameters within the Earth–atmosphere interface
covering forest, wetland and lake areas (Hari and Kulmala, 2005). Furthermore, SMEAR II is part of large research
networks such as Aerosols, Clouds and TRace gases InfraStructure (ACTRIS), Integrated Carbon Observation System
(ICOS), Europe's Long–term Ecosystem Research (LTER) and the infrastructure for Analysis and Experimentation on
Ecosystems (AnaEE).

The chemical composition of aerosol particles at SMEAR II has been studied previously in multiple short (<1 – 10 month)
measurement campaigns with both offline (Saarikoski et al., 2005;Kourtchev et al., 2005;Cavalli et al., 2006;Finessi et
al., 2012;Corrigan et al., 2013;Kourtchev et al., 2013;Kortelainen et al., 2017), and online methods (Allan et al.,
2006;Finessi et al., 2012;Häkkinen et al., 2012;Corrigan et al., 2013;Crippa et al., 2014;Hong et al., 2014;Makkonen et
al., 2014;Äijälä et al., 2017;Hong et al., 2017;Kortelainen et al., 2017;Riva et al., 2019;Äijälä et al., 2019). The previous
studies include several important discoveries regarding SMEAR II aerosol composition. For example, a large mass
fraction of particulate matter has been found to be (oxidised) organic aerosol (Corrigan et al., 2013;Crippa et al.,
2014;Äijälä et al., 2017;Äijälä et al., 2019), and recognised as terpene oxidation products (Kourtchev et al., 2005;Allan
et al., 2006;Cavalli et al., 2006;Finessi et al., 2012;Corrigan et al., 2013;Kourtchev et al., 2013). In addition to this forest-
generated SOA, a nearby sawmill also contributes significantly to the OA loading in the case of south easterly winds (e.g.
Liao et al., 2011;Corrigan et al., 2013;Äijälä et al., 2017). The composition of the sawmill-OA is found to significantly
resemble biogenic OA (Äijälä et al., 2017). Also biomass burning organic aerosol (BBOA) contributes to the OA mass
(Corrigan et al., 2013; Crippa et al., 2014; Äijälä et al., 2017; Äijälä et al., 2019). The BBOA presence in the summertime
aerosol depends on long-range transport of wildfire plumes. Inorganic species from anthropogenic activities, majority
identified as ammonium sulphate and nitrate, are also transported to the station. Ammonium sulphate represents the
dominating inorganic species (e.g. Saarikoski et al., 2005; Äijälä et al., 2019). Despite the long list of studies and
discoveries, the measurement/sampling periods have occurred mostly between early spring and late autumn, leaving the
description of the wintertime aerosol composition nearly fully lacking. Hence, based on these studies alone, the
understanding of the degree of variability in aerosol chemical composition at SMEAR II is incomplete – in both intra-
and inter-annual scales.
Here, we provide a comprehensive overview of sub-micron aerosol chemical composition at SMEAR II. This study does
not only provide the analysis of the longest time series of sub–micron aerosol chemical composition measured on–line in
the climate–sensitive boreal environment, but also introduces the data set to the scientific community for further utilisation
with other SMEAR II, ACTRIS, ICOS, LTER and AnaEE data to improve our understanding of the aerosol sensitivity of
the (boreal) climate.
**2 Measurements and methods**
In this chapter, we introduce the SMEAR II measurement site, data processing and analysis tools. As the meteorological
conditions ruling at the station are of high importance influencing sub-micron aerosol chemical composition, we will first
focus on giving an overview of the SMEAR II climate. The instrument operation, data processing and analysis part briefly
describes the instrumentation used, focusing mainly on the aerosol chemical speciation monitor (ACSM) that serves as
the key instrument for the current study.
**2.1 SMEAR II description**
The measurements reported here were conducted at the SMEAR II station (61°51'N, 24°17'E, 181 m above sea level)
(Hari and Kulmala, 2005) between years 2012 and 2018. However, they continue also after 2018 as part of the station's
long-term measurements. SMEAR II is located in a nearly 60-year-old Scots pine (Pinus sylvestris) dominated stand. A
previous land use survey reveals that a majority of the area surrounding the station is forested, as 80% of the land within
5 km radius and 65% within 50 km radius is covered by mixed forest (Williams et al., 2011). The forested area located
north west of the station is shown to have least anthropogenic air pollutant sources (Williams et al., 2011;Tunved et al.,
2006). However, 90% of the forests in Fennoscandia region are introduced to anthropogenic influence via forest
management (Gauthier et al., 2015). The city of Tampere (population approximately 235 000) lies within the 50 km radius

to the south west introducing a notable source of anthropogenic pollution. Other evident nearby sources of anthropogenic pollution are the town of Orivesi (population approximately 9 200) 19 km south of SMEAR II and the nearby village of Korkeakoski with two saw mills and a pellet factory 6–7 km to the south east from SMEAR II (Liao et al., 2011;Äijälä et al., 2017). Nonetheless, the dominating source of air pollutants are air masses advected from industrialized areas over southern Finland, St. Petersburg region in Russia and continental Europe (Kulmala et al., 2000;Patokoski et al., 2015;Riuttanen et al., 2013). The anthropogenic emissions are minor at the station. Monoterpenes, notably α–pinene and $\Delta^3$–carene, are the dominating emitted biogenic non–methane VOCs from the forest (Hakola et al., 2012;Barreira et al., 2017).

The mean annual temperature at the measurement station during the measurement period (2012–2018), recorded at 4.2 m above ground level, was 5.4 °C. In average, January was the coldest month ($T_{avg}$ = –6.2 °C) and July the warmest ($T_{avg}$ = 16.6 °C). The mean annual temperature recorded was ca. 2°C higher than the 1981–2010 annual mean reported (Pirinen et al., 2012). The seasonal and diurnal variation of temperature is presented in Figure 1a followed by the corresponding data for global radiation above the forest (Figure 1b). November and December were the darkest months whereas the radiation maximum was reached in late May and early June, which is earlier than generally observed at the top of the atmosphere. The reason for the early radiation maximum peaking time is the increased fractional cloud cover in July (Tuononen et al., 2019), likely promoted by convection. The formation of convective clouds hinders the transmission of solar radiation to the lower troposphere, and increases the intensity of precipitation, making the 1981–2010 mean precipitation maximum occur in July (92 mm) (Pirinen et al., 2012). However, November, December and January hold the greatest amount of precipitation days (≥ 0.1 mm for ca. 21 days per month). The annual 1981–2010 mean cumulative precipitation is 711 mm (Pirinen et al., 2012). The first snow on the ground can be expected in November, and the snow depth maximum is commonly reached in March. The snow cover is roughly lost in April (Pirinen et al., 2012). The wind direction recorded above the forest canopy during the measurement period is normally from the south west with enhanced southerly influence during winter months (especially in January and February) and has large scatter during summer months (Figure 1c&Figure A.1a-c). The diurnal mean of wind speeds above the forest canopy were usually greatest during wintertime as can be expected based on overall Northern hemispheric behaviour. The seasonal cycles of wind speed show most diurnal variability from May to September (Figure 1d&Figure A.1a-c).

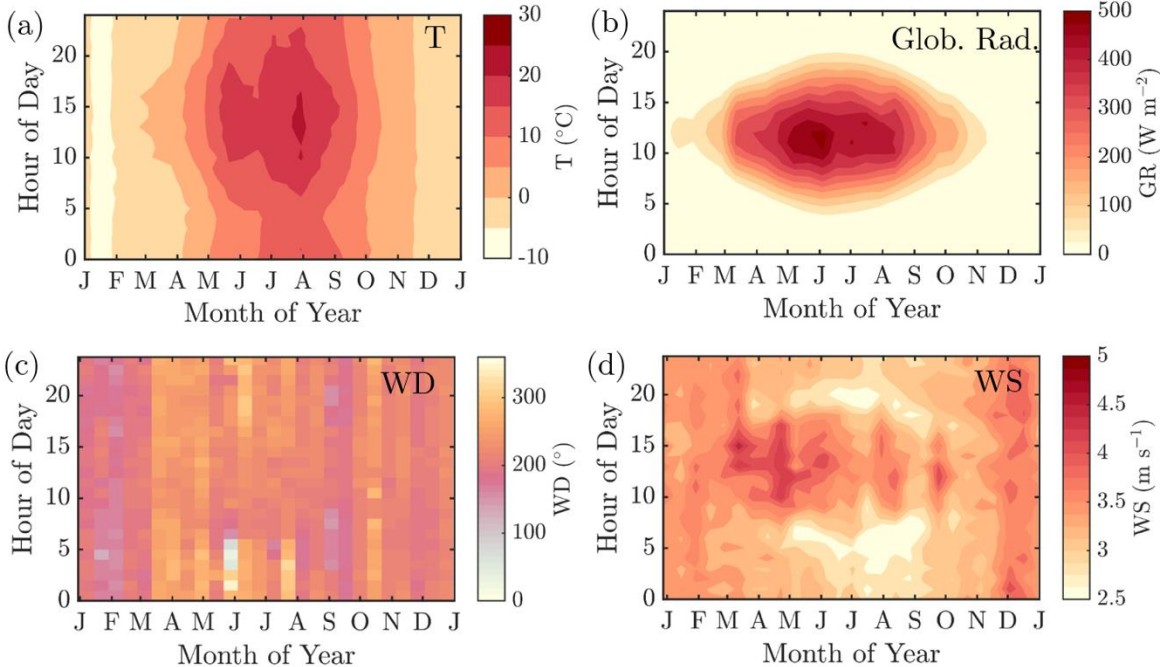

184

**Figure 1** The seasonal evolution of diurnal cycles of ambient temperature measured 4.2 m above ground level (panel a), global radiation above the forest canopy (panel b), wind direction above the forest canopy (panel c), and wind speed above the forest canopy (panel d) recorded at SMEAR II station in 2012 – 2018. The y–axes in the figures represent the local time of day (UTC+2) and the x–axes the time of the year. The colour scales correspond to the temperature in degrees Celsius, global radiation in W m$^{-2}$, wind direction in degrees and wind speed in m s$^{-1}$, respectively. Panels a, b and d include interpolation of the 14 d × 1 h resolution data grid into isolines based on the MATLAB2017a *contourf* function. Panel c has no interpolation involved due to challenges related to interpolating over a circularly behaving variable. The plot is produced with MATLAB 2017a *pcolor* function.

192

**2.2 ACSM measurements**

The Aerosol Chemical Speciation Monitor (ACSM; Aerodyne Research Inc. USA) was first described by Ng et al. in 2011. It was developed based on the Aerosol Mass Spectrometer (AMS) (Canagaratna et al., 2007), but simplified at the cost of mass and time resolution to achieve a robust instrument for long–term measurements. The ACSM samples ambient air with a flow rate of 1.4 cm$^3$ s$^{-1}$ through a critical orifice (100 µm in diameter) towards an aerodynamic lens efficiently transmitting particles between approximately 75 and 650 nm in vacuum aerodynamic diameter ($D_{va}$) and pass through particles further up to 1µm in $D_{va}$ with a less efficient transmission (Liu et al., 2007). After this, the particles are flash vaporized at a 600 °C hot surface in high vacuum and ionised with electrons from a tungsten filament (70 eV, electron impact ionisation, EI). These processes lead to substantial fragmentation of the molecules forming the aerosol particles. The resulting ions are guided to a mass analyser, a residual gas analyser (RGA) quadrupole, scanning through different mass–to–charge ratios ($m/Q$). The detector is a secondary electron multiplier (SEM). The particulate matter detected by the ACSM is referred to as non–refractory (NR) sub–micron particulate matter (PM$_1$). The word 'non–refractory' (NR) is attributed to the instrument limitation to detect material flash evaporating at 600 °C thus being unable to measure heat–resistant material such as minerals or soot. The word 'PM$_1$' is linked to the aerodynamic lens approximate cut-off at 1 µm. Importantly, the NR–PM$_1$ reported from these ACSM measurements is a difference between the signal of particle-laden air and signal recorded when the sampling flow passed a particle filter (filtered air).


The ACSM measurements for the current study were conducted within the forest canopy through the roof of an air
conditioned container. A PM$_{2.5}$ cyclone was used to filter out big particles that could cause clogging of the critical orifice.
A Nafion dryer was installed in 2013 upstream the instrument ensuring a sampling relative humidity (RH) below 30%.
Before this, the RH was not controlled nor recorded. Thus, the RH was likely high during summer, but low during
wintertime. Moreover, a 3 litres per minute (Lpm) overflow, which was ejected only before the aerodynamic lens, was
used to minimise losses in the sampling line (length approximately 3 m). The data were acquired using the ACSM data
acquisition software (DAQ) provided by Aerodyne Research Inc., the instrument manufacturer. The DAQ version was
updated upon new releases. The ACSM was operated to perform $m/Q$ scans with a 200 ms Th$^{-1}$ scan rate in the mass–to–
charge range of $m/Q$ 10 Th to 140 Th. Filtered and particle–laden air were measured interchangeably for 28 quadrupole
scans resulting in ca. 30 minute averages. The air signal, obtained from the automatic filter measurements, was subtracted
from the sample raw signal, yielding the signal from aerosol mass only. The data processing was performed using ACSM
Local v. 1.6.0.3 toolkit within the Igor Pro v. 6.37 (Wavemetrics Inc., USA). Upon data processing, the different detected
ions were assigned into organic or inorganic species bins (i.e. total organics, sulphate, nitrate, ammonium and chloride)
using a fragmentation table (Allan et al., 2004). Moreover, the data were normalized to account for N$_2$ signal variations
related to ACSM flow rate and sensitivity changes (due to SEM voltage response decay).

The ACSM raw signal (IC) is converted to mass concentration ($C$) with the following equation obtained from Ng et al.

227  (2011):

$$C_{\mathrm{s}} = \frac{1}{\mathrm{CE} \times \frac{T_m}{Q}} \times \frac{10^{12}}{\mathrm{RIE_s} \times \mathrm{RF_{NO_3}}} \times \frac{Q_{\mathrm{cal}} \times G_{\mathrm{cal}}}{Q \times G} \times \sum_{i=0}^{n} \mathrm{IC}_{s,i}, \tag{1}$$

where $C_s$ is the concentration of species $s$, CE is the particle collection efficiency (see chapter 2.4 ACSM collection
efficiency correction), and $T_{m/Q}$ the $m/Q$ –dependent ion transmission efficiency in the RGA quadrupole mass analyser.
The $T_{m/Q}$ is constantly recorded based on naphthalene fragmentation patterns and their comparison to naphthalene
fragmentation pattern in the NIST data base (75 eV EI; http://webbook.nist.gov/). Naphthalene is used as an internal
standard in the ACSM and is thus always present in the mass spectrum (Ng et al., 2011). The RIE$_s$ is the relative ionisation
efficiency of species $s$ and RF$_{NO_3}$ the ACSM response factor determined through ionisation efficiency (IE) calibrations
with ammonium nitrate (NH$_4$NO$_3$). The RF$_{NO_3}$ explains the ACSM ion signal (A) per µg m$^{-3}$ of nitrate. $Q_{cal}$ and $G_{cal}$ are
the ACSM volumetric flow rate and detector gain during ACSM calibration, whereas $Q$ and $G$ are the values during the
measurement period for volumetric flow rate and detector gain, respectively. They generally correspond the calibration
values. The final parameter is the sum of the signal introduced by individual ions (IC$_{s,i}$) originating from species $s$.

The ionisation efficiency calibration was performed with dried and size–selected ammonium nitrate (NH$_4$NO$_3$) particles
to retrieve the RF$_{NO_3}$ parameter required in Equation (1). In addition, ammonium sulphate ((NH$_4$)$_2$SO$_4$) calibrations were
carried out, albeit less frequently, providing a value for the sulphate relative ionization efficiency (RIE $_{SO_4}$). RIE$_{NH_4}$ was
derived from the ammonium nitrate calibration. Constant RF and RIE values were used for each year, respectively. The
conversion from ACSM raw signal to mass concentration was performed with the ACSM Local v. 1.6.0.3 provided by
Aerodyne Research Inc.

**2.3. Additional measurements**

In addition to the ACSM–measurements, the SMEAR II station has a large number of other air composition related measurements. In the current study, we investigate only a small fraction of them. The particle measurements (i.e. ACSM, Differential Mobility Particle Sizer (DMPS), Dekati cascade impactor, Aethalometer, Organic Carbon and Elemental Carbon (OCEC) analyser and Monitor for AeRosols and Gases in ambient Air (MARGA) 2S) were conducted in two measurement containers and a cabin, all located within ca. 50 m from each other. The gas phase sampling of $NO_x$, $SO_2$, CO and VOCs as well as temperature, wind and global radiation measurements, were conducted from the station mast that has several measurement heights from near ground to 127 m height. The temperature measurement was conducted with a Pt100 sensor (4.2 m above ground level), the global shortwave radiation with a Middleton SK08 pyranometer (125 m above ground level). The horizontal wind measurements were conducted with Thies 2D Ultrasonic anemometers above the forest canopy (16.8 to 67.2 m above ground level). The $NO_x$ measurements were performed with a TEI 42 iTL chemiluminescence analyser equipped with a photolytic Blue Light Converter ($NO_2^{\bullet}$ to $NO^{\bullet}$) converter, $SO_2$ with a TEI 43 iTLE fluorescence analyser, and CO with IR absorption analysers Horiba APMA 370 (until January 2016) and API 300EU (from February 2016 onwards). The $NO_x$, $SO_2$, CO and PTR–MS sampling was conducted 67.2 m above ground level. The $SO_2$, $NO_x$, CO, and PTR–MS (only 2012–2013), and meteorology data were uploaded from Smart–SMEAR data base (https://avaa.tdata.fi/web/smart) (Junninen et al., 2009). The DMPS, Dekati impactor, Aethalometer, OCEC–analyser and MARGA–2S measurements were conducted within the forest canopy are described in more detail in the sections below. The data availability is shown in Figure A.2.

**2.3.1 DMPS**

The Differential Mobility Particle Sizer (DMPS) measures the aerosol size distribution below 1 µm electrical mobility diameter. SMEAR II holds the world record in online aerosol size distribution measurements (Dada et al., 2017), as the measurements started already in 1996. The DMPS system is described in detail previously (Aalto et al., 2001). Briefly, the SMEAR II DMPS is a twin DMPS setup that samples 8 m above ground from an inlet with a flow rate of 150 Lpm. The measurement cycle is 10 minutes. The first DMPS (DMPS–1) has a 10.9 cm long Vienna type differential mobility analyser (DMA) and a model TSI3025 condensation particle counter (CPC) that was changed to model TSI3776 after October 2016. The sheath flow rate in the DMA is 20 Lpm and aerosol flow rate 4.0 Lpm. The measurement range of the DMPS–1 is 3–40 nm. The second DMPS (DMPS–2) has a 28 cm long Vienna type DMA and a TSI3772 CPC. The sheath flow rate is 5 Lpm and the aerosol flow rate 1 Lpm. The measurement range is 20 nm – 1 µm. The sheath flows are dried (RH < 40%), and controlled with regulating valves as well as measured with TSI mass flow meters operated in volumetric flow mode. The aerosol flow is brought to charge balance with 370 MBq C–14, and after March 2018 with a 370 MBq Ni–63 radioactive beta source. The aerosol flow rates are monitored with pressure drop flow meters. The aerosol flows were not dried. Temperatures and RHs are monitored from DMPS excess flow and from the aerosol inlet. The aerosol flow rates were checked and adjusted every week against a Gilian Gilibrator flow meter throughout the measurement period. The DMA high voltages were also validated with a multimeter. The CPC concentrations were compared against each other with size–selected ammonium sulphate particles in the 6–40 nm range as well as compared against the TSI3775 particle counter that measures the total aerosol particle number concentration at the station. The sizing accuracy of the two DMAs were cross-compared with 20 nm ammonium sulphate particles. In addition, the accuracies of the RH, temperature and pressure probes were validated each year.

### 2.3.2 Cascade impactor

The $PM_1$ and $PM_{2.5}$ (particulate mass of aerosol particles with an aerodynamic diameter below 2.5 µm) mass concentrations measured between 2012–2017, which were included in the current study, were retrieved from the cascade impactor measurements. This gravimetric $PM_{10}$ impactor, produced by Dekati Ltd., is a three–stage impactor with cut–points at 10, 2.5 and 1 µm. The collection is conducted on greased (Apiezon vacuum grease diluted in toluene) Nuclepore 800 203 25 mm polycarbonate membranes with 30 Lpm flow rate, approximately 5 m above ground level. The filter smearing was performed to avoid losses due to particle bouncing. The filters were weighed manually every 2–3 days and stored in a freezer for possible further analysis.

### 2.3.3 PTR–MS

The monoterpene concentration was measured using the proton transfer reaction quadrupole mass spectrometer (PTR–MS) manufactured by Ionicon Analytik GmbH, Innsbruck, Austria  (Lindinger and Jordan, 1998). The monoterpene measurement setup is described in detail previously (Rantala et al., 2015). Shortly, the PTR–MS was placed inside a measurement cabin on the ground level and the sample air was drawn down from a measurement mast to the instrument using a 157 m long PTFE tubing (16/14 mm o.d./i.d.). The sampling line was heated and the sample flow was 45 Lpm. However, the sample entering the PTR–MS was only 0.1 Lpm. During the study period, the primary ion signal $H_3O^+$ (measured at isotope $m/Q$ 21 Th) varied slightly around $5–30 \times 10^6$ c.p.s. (counts per seconds). The instrument was calibrated every 2–4 weeks using three different VOC standards (Aper–Riemer) and the instrumental background was measured every third hour using VOC free air, produced by a zero air generator (Parker ChromGas, model 3501). Normalised sensitivities and the volume mixing ratios were then calculated using the method introduced previously (Taipale et al., 2008). For example, the normalized sensitivity of alpha–pinene (measured at $m/Q$ 137 Th) varied between 2 and 5 n.c.p.s. p.p.b.$^{-1}$ over the study period. Only the signal of monoterpenes at $m/Q$ 137 Th were analysed in the current study.

### 2.3.4 Aethalometer

The concentration of equivalent black carbon (eBC) in the $PM_1$ size range was measured by using two different Magee Scientific Aethalometer models: AE–31 during 2012–2017, and AE–33 in 2018. The sample air was taken through an inlet equipped with a $PM_{10}$ cyclone and a Nafion dryer, and a $PM_1$ impactor. Aethalometers determine the concentration of eBC by collecting aerosols on a filter medium and measuring the change in light attenuation trough the filter. Both of the Aethalometers quantify eBC concentration optically at seven wavelengths (370, 470, 520, 590, 660, 880 and 950 nm). Only the eBC concetration determined at 880 nm was used in the current study. AE–31 data was corrected for a filter loading error with a correction algorithm derived previously (Collaud Coen et al., 2013). A mass absorption cross section of 4.78 $m^2$ $g^{-1}$ at 880 nm was used in the eBC concentration calculation. The AE–33 used a "dual–spot" correction is described previously (Drinovec et al., 2015).

### 2.3.5 OCEC–analyser

Organic carbon (OC) and elemental carbon (EC) concentrations were measured using a semi–continuous Sunset OCEC analyser (Bauer et al., 2009) produced by Sunset Laboratories Inc. (USA). The aerosol sampling was conducted through the same container roof as the ACSM. The inlet length was approximately the same as for the ACSM (ca. 3 m). The sample flow was guided through a $PM_{2.5}$ cyclone and a carbon plate denuder to avoid collection of large particles and a

positive artefact introduced by organic vapours. In the OCEC, the sample is collected on a quartz–filter for 2.5 hours with an 8 Lpm flow rate. The sampling procedure is followed by the analysis phase. The analysis phase includes thermal desorption of PM from the filter following the EUSAAR–2 protocol (Cavalli et al., 2010), and introducing the aerosol sample to inert helium gas that is used to carry the OC to a $MnO_2$ oxidising oven. This leads to OC oxidation to $CO_2$, which is then quantified, with a non–dispersive infrared (NDIR) detector. Afterwards the remaining sample is introduced to a mixture of oxygen and helium enabling EC transfer to the oven. The resulting $CO_2$ from EC desorption and combustion is also quantified using the NDIR detector. An additional optical correction was used to account for the amount of pyrolysed OC during the helium phase. EC was also quantified using a laser installed in the analyser. This method is similar to the Aethalometer (see chapter 2.3.4 Aethalometer). After each analysis phase, a calibration cycle was performed via methane oxidation. The instrument was maintained extensively in November 2017, thus making the year 2018 most reliable for ACSM comparison. Only data measured in 2018 was used in this study.

**2.3.6 MARGA–2S**

Inorganic gases (HCl, $HNO_3$, HONO, $NH_3$, $SO_2$) and major inorganic ions in $PM_{2.5}$ and $PM_{10}$ fraction ($Cl^-$, $NO_3^-$, $SO_4^{2-}$, $NH_4^+$, $Na^+$, $K^+$, $Mg^{2+}$, $Ca^{2+}$) were measured with one hour time–resolution using the online ion chromatograph MARGA 2S ADI 2080 (Applikon Analytical BV, Netherlands). In the MARGA instrument, ambient air was taken through the inlet to a wet rotating denuder where the gases were diffused in absorption solution (10 p.p.m hydrogen peroxide). Aerosol particles that passed through denuder were collected in a steam jet aerosol collector. The sample solutions from the denuder as well as the steam jet aerosol collector were collected in syringes and injected in an anion and cation ion chromatograph with an internal standard solution (LiBr). Cations were separated in a Metrosep C4 (100/4.0) cation column using 3.2 mmol $L^{-1}$ MSA eluent. For anions, a Metrosep A Supp 10 (75/4.0) column with $Na_2CO_3$ – $NaHCO_3$ (7 mmol $L^{-1}$ / 8 mmol $L^{-1}$) eluent were used. The detection limits for all the components were 0.1 µg $m^{-3}$, or smaller. The unit used for the current study is described in more detail previously (Makkonen et al., 2012).

**2.4 ACSM collection efficiency correction**

The ACSM data processing includes correcting for the measurement collection efficiency (CE) that is estimated to be approximately 0.45–0.5 in average for AMS-type instruments (Middlebrook et al., 2012). The reduction is caused by particle bouncing at the instrument vaporizer (Middlebrook et al., 2012). Middlebrook et al. (2012) provide a method to estimate the CE, based on aerosol chemical composition. However, this method was not applicable to our data set due to low, and thus noisy, ammonium signals that were most of the time near the instrument detection limit. Thus, we chose to calculate the collection efficiency based on the ratio between the NR–$PM_1$ (total mass concentration measured by the ACSM) and a Differential Mobility Particle Sizer (DMPS)–derived mass concentration (after subtracting the equivalent black carbon, eBC). The DMPS-derived mass concentration is determined as follows: 1) Calculation of aerosol volume concentration ($m^3$ $m^{-3}$) of the ACSM detectable size range, where the aerodynamics lens transmission is most efficient (50–450 nm in electrical mobility = ~75–650 nm in vacuum aerodynamic diameter) and assuming spherical particles, 2) Estimating aerosol density based on the ACSM-measured chemical composition ($\rho_{(NH_4)_2SO_4}$= 1.77 g $cm^{-3}$, $\rho_{NH_4NO_3}$= 1.72 g $cm^{-3}$, $\rho_{Org}$= 1.50 g $cm^{-3}$, $\rho_{BC}$ = 1.00 g $cm^{-3}$), 3) Calculating the mass concentration (µg $m^{-3}$; mass concentration = density × volume concentration). As direct scaling of ACSM data to the DMPS-derived and eBC subtracted mass concentration is strongly not recommended, we chose to use two-month running medians of the ratio between the NR–$PM_1$ and eBC-subtracted DMPS-derived mass concentration. The two–month running median approach diminishes the effect instrument

noise in the DMPS–derived mass concentration that could otherwise be introduced as additional uncertainty into the
ACSM–data. The two-month median CEs were within 10% of the annual mean values in years 2013–2018. In 2012 the
CE had stronger seasonal variation (16% variation around the mean, peaking in summer) likely due to the lack of the
aerosol dryer in the sampling line. The magnitudes of the CEs can be obtained from Figure 2a.

Figure 2a depicts the linear regression fits for ACSM mass concentration (without CE correction) and eBC–subtracted
DMPS–derived mass concentration scatter plots for each year. The correlation coefficients (Pearson $R^2$) between these
two independently measured variables are high, indicating that both ACSM and DMPS functioned well throughout the
long dataset. Years 2012, 2016–2018 have linear regression fit slopes ($k$) corresponding to CE values reported in the
literature (Middlebrook et al., 2012), whereas slopes for years 2013–2015 were higher than expected. The most likely
reason for these high values were calibration difficulties that might have led to underestimation of the instrument $RF_{NO_3}$
that is required in the mass concentration calculation. This possible $RF_{NO_3}$ underestimation was accounted for in the
DMPS–based CE correction with higher CE values than theoretically suggested for ACSM–systems. The resulting
agreement between the eBC–subtracted DMPS–derived mass concentration and the CE corrected ACSM–derived mass
concentration is presented in Figure 2b. As the NR–PM$_1$ incorporates the two–month running median of the CE calculated
using DMPS–data, it is not surprising that a good correlation was achieved.

Figures 2c&d visualise the relationship between ACSM–derived mass concentration and a Dekati impactor PM$_1$ data
(see 2.3.2 Cascade impactor for instrument description) before and after CE correction, respectively. The impactor PM$_1$
is not eBC subtracted as it would have significantly decreased the number of points in the analysis. The degree of
agreement between the ACSM and impactor measurements is significantly lower compared to the agreement between the
ACSM and DMPS. The reason for this is likely the fact that the Dekati impactor measurements are prone to uncertainties
due to long sampling times and manual weighing. This scatter is reduced slightly in Figure 2d compared to Figure 2c due
to the DMPS–based CE correction and the slope–values ($k$) increase. As the agreement between the ACSM–derived mass
concentration and impactor PM$_1$ is better after CE correction both due to increased correlation coefficients ($R^2$) and slopes
($k$), the (two month running median) DMPS–based CE correction is justified. Hereafter, all the ACSM data presented and
discussed are CE corrected. We refer to it as NR–PM$_1$. The CE correction method applied importantly also ensures more
quantitative year-to-year comparability of the ACSM data acquired as it also corrected for the overestimated calibration
values obtained during 2013–2015.
**2.5 ACSM chemical speciation validation**
To validate the ACSM chemical speciation process, the ACSM organics (Org) and sulphate (SO$_4$) were compared against
the organic carbon (OC) measured by a Sunset OCEC–analyser (see section 2.3.5 OCEC-analyser for instrument) (Figure
2e) and water-soluble sulphate measured by a MARGA–2S (see section 2.3.5 MARGA-2S for instrument description)
(Figure 2f), respectively. Both of these reference instruments sample the PM$_{2.5}$ range, and thus we expect them to detect
also larger particles and thus a higher mass loading than the ACSM, when particles exceeding 1 µm in aerodynamic
diameter are present in the ambient air. The OC and Org measurements show a high degree of agreement indicated by
Pearson $R^2$ of 0.72 during the overlapping measurement period at SMEAR II. The slope of the linear regression fit ($k$ =
1.52) is comparable to literature values of organic matter to organic carbon ratios (OM:OC) (Turpin and Lim, 2001;Lim
and Turpin, 2002;Russell, 2003).  The linear regression is calculated using all the overlapping data from year 2018, when
the OCEC was well functioning after instrument service. Regarding the water-soluble inorganic ions, only the $SO_4^{2-}$
concentration (in $PM_{2.5}$), retrieved from the MARGA measurements, was used for the current analysis for ACSM data
validation purposes. The nitrate time series, for example, are known to be different between the two measurements at
SMEAR II (Makkonen et al., 2014). The scatter between the nitrate measurements, visualised also here, in Figure A.3,
could serve as evidence of organic nitrates, which are not efficiently detected by MARGA (Makkonen et al., 2014). The
presence of organic nitrates are discussed later in the manuscript (see chapter 3.2 Diurnal variation of NR–$PM_1$
composition). Other factors influencing the nitrate agreement could arise from the MARGA nitrate background
subtraction procedure, the overall low nitrate signal at SMEAR II (Makkonen et al., 2014), an organic $CH_2O^+$ fragment
coinciding with $NO^+$ at $m/Q$ 30 Th leading to ACSM nitrate over prediction under certain conditions, and the difference
between the ACSM  and MARGA size cuts ($PM_1$ vs $PM_{2.5}$). The ACSM sulphate, however, correlates well with MARGA
(Pearson $R^2 = 0.77$), but has a slightly lower Pearson $R^2$ compared to an earlier $< 11$–month MARGA vs AMS comparison
from SMEAR II (Pearson $R^2 = 0.91$) (Makkonen et al., 2014). Overall, based on the good agreement between ACSM and
Sunset OCEC, MARGA, DMPS and Dekati cascade impactor measurements, we are confident of the year–to–year
comparability of our ACSM dataset.

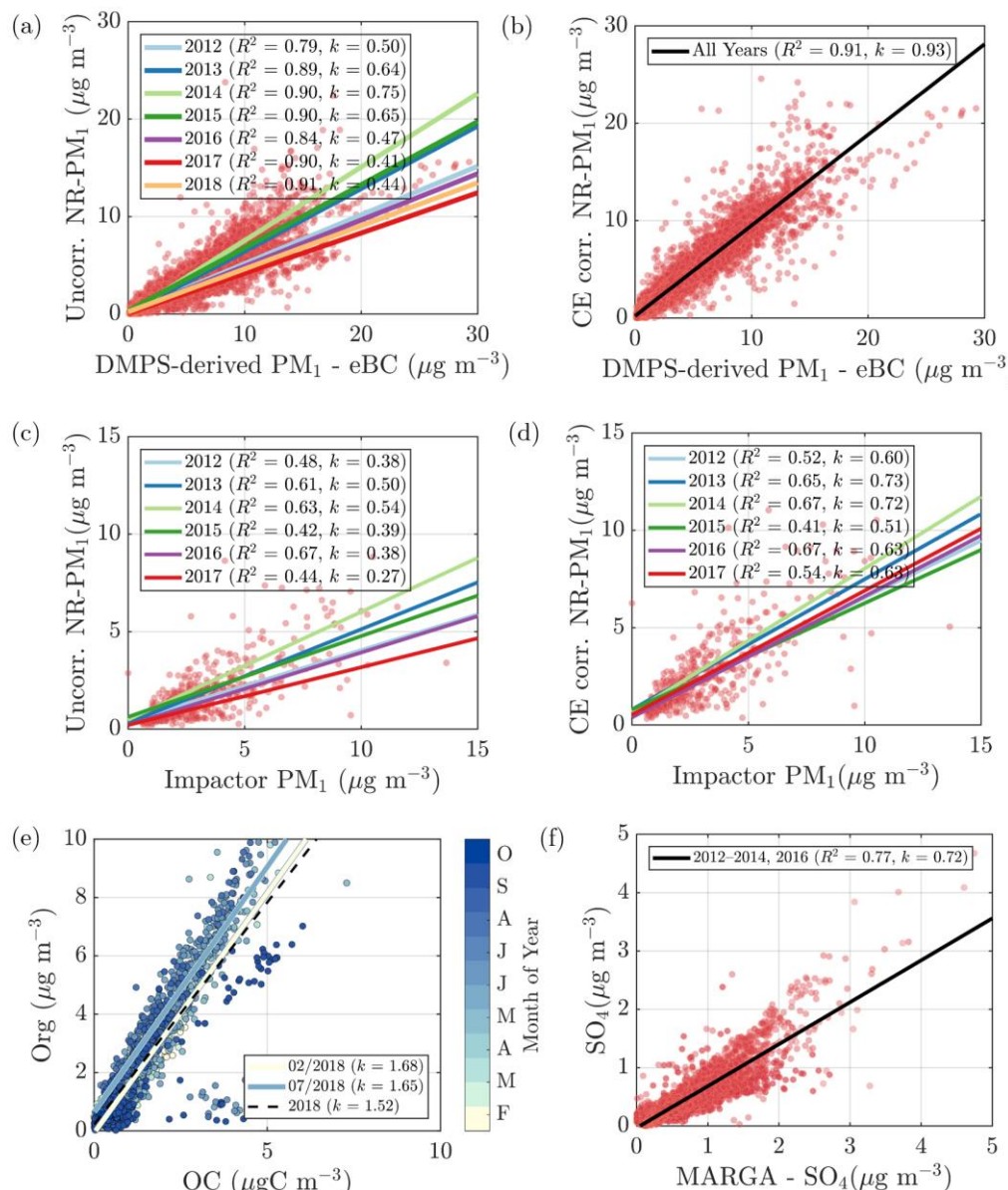


**Figure 2** (Panel a) The NR–PM$_1$ mass concentration without collection efficiency (CE) correction vs the DMPS–derived eBC subtracted mass concentration. The linear fits are displayed with solid lines for each year, respectively. The slopes of the linear fits ($k$) and Pearson correlation coefficients ($R^2$) are presented in the figure legend. (Panel b) CE–corrected NR–PM$_1$ mass concentration vs the DMPS–derived eBC subtracted mass concentration. The black line represents the overall linear fit. (Panel c) The NR–PM$_1$ mass concentration without collection efficiency (CE) correction vs the Dekati impactor PM$_1$ concentration. The linear fits are displayed with solid lines for each year, respectively. (Panel d) CE–corrected NR–PM$_1$ mass concentration vs correction vs the Dekati impactor PM$_1$ concentration. The linear fits are displayed with solid lines for each year, respectively. (Panel e) The ACSM organics vs the PM$_{2.5}$ organic carbon (OC) detected with a semi–continuous OC/EC analyser. The black dashed line represents the overall linear fit (Pearson $R^2 = 0.72$), and the solid white and blue linear fit lines represent the February and July fits, respectively. The color coding indicates the month of the year in 2018. (Panel f) The ACSM sulphate vs the PM$_{2.5}$ sulphate detected with MARGA–2S. The black line represents the overall linear fit. In all of the panels, red dots represent all the measurement points collected in the course of the measurement period.

428

**2.6 Openair polar plots with ZeFir pollution tracker**

The wind direction dependence of different NR–PM$_1$ chemical species observed at SMEAR II were investigated with openair bivariate polar plots. Openair is an open source, R–based package described previously (Carslaw and Ropkins, 2012). Briefly, the openair polar plots show how the pollutant concentration varies under different wind speed and direction. The calculation of the polar plots are based on binning pollutant concentration data into different wind direction and speed bins followed by the concentration field interpolation. As the polar plots do not take the frequencies of the wind direction nor wind speed into account, they should be investigated together with a traditional wind rose representing the likelihood of each wind direction and wind speed combination. The ZeFir pollution tracker (Petit et al., 2017), an Igor Pro (Wavemetrics Inc, USA) based graphical interface for producing openair polar plots among other functionalities, was utilized in the current study. Median statistics with fine resolution were set in the ZeFir–based openair initialisation. These plots provide an informative first step for tracking PM$_1$ and its precursors' wind direction and speed dependence.

**3 Results and discussion**

First, we state that the ACSM data set was not long enough to provide sufficient statistics for investigation of long–term trends of NR–PM$_1$ or its components' loading, hence no analysis of such is presented here. In this section we discuss the inter– and intra–annual variation in sub–micron non–refractory aerosol chemical composition at SMEAR II in 2012–2018. We first introduce the monthly scale behaviour and year–to–year variability. We briefly introduce two case studies, one linked to elevated sulphate loading at the station due to a lava field eruption in Iceland, and another one discussing the effect of heatwaves on PM$_1$ loading and composition. Hereafter, we introduce the overall median diurnal profiles of individual chemical species observed in the NR–PM$_1$, and finally the chemical composition observations linked to wind speed and direction observations above the forest canopy.

**3.1 Inter– and intra–annual variation**

The monthly median seasonal cycles of NR–PM$_1$ and PM$_{2.5}$ show bimodal distributions as the PM loading has two maxima: one peak in February, and another one in summer (June, July, and August), the latter one being more significant. This can be observed from Figures 3a&b, where the monthly median PM loading for each year is visualised. The NR–PM$_1$ seasonal cycle (Figure 3a) is more pronounced compared to the PM$_{2.5}$ cycle (Figure 3b). A possible reason for this could be the lack of PM$_{2.5}$ data in 2018 that is having a high impact on the NR–PM$_1$ July peak. The PM$_1$/PM$_{2.5}$ ratio (Figure 3c) in turn, calculated using the Dekati Impactor data alone, demonstrates that most of the time, 60–80% of PM$_{2.5}$ can be explained by PM$_1$. The ratio is lowest (60–70%) in wintertime (December, January, February) implying an increased mass fraction of particles with aerodynamic diameters greater than a micrometre, compared to the summertime, when the ratio is nearly 80%.

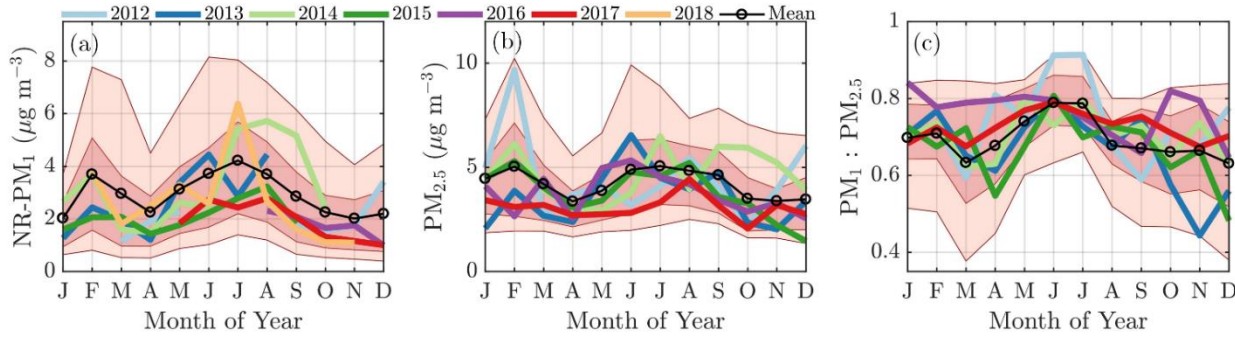

**Figure 3** The monthly cycles of NR–PM$_1$ (Panel a), PM$_{2.5}$ (Panel b), and the ratio between PM$_1$ and PM$_{2.5}$ (Panel c). The median monthly values for each year are individually displayed with the coloured solid lines. The black circled line represents the overall monthly mean values. The dark red shaded area is drawn between the overall 25$^{th}$ and 75$^{th}$ percentiles and the lighter red shaded area between the overall 10$^{th}$ and 90$^{th}$ percentile. The x–axes represent the time of the year and the y–axes in panels a and b mass concentration in µg m$^{-3}$ and a unit less ratio in panel c.

The February PM maximum is linked to an enhanced loading of inorganic aerosol species, such as nitrate and especially sulphate, as well as a slight increase of organics (Figures 4a-c), whereas the summertime maximum is explained by a massive enhancement of organics alone (Figure 4a). The main precursors for inorganic aerosols, i.e. SO$_2$ and NO$_x$ peak during winter albeit less sharply on February alone (Figure 4e&f). As fossil fuel combustion processes are the major sources of SO$_2$ and NO$_x$, their emissions likely increase during cold months due to enhanced need for residential heating, for example. More importantly, these emissions are trapped in a shallow atmospheric boundary layer increasing the concentration recorded within it. One possible reason for the "lack" of sulphate and nitrate aerosol outside February, despite the great availability of SO$_2$ and NO$_x$, is likely related to wind direction transitioning, discussed later in the manuscript. Another effect could be the darkness prohibiting photochemistry needed for inorganic aerosol formation. Indeed, the global radiation measured above the forest canopy shows only a minimal short wave radiation flux in November, December and early January, but it increases mid–January onwards (Figure 1b). As February is generally drier (in terms of less precipitation) than the other winter months (Pirinen et al., 2012), the lifetime of aerosols could be greater, making the inorganic particles more likely to reach SMEAR II. The inorganic nitrate (ammonium nitrate) formation is highly dependent on ammonia availability. The ammonia concentration during wintertime is nearly negligible at SMEAR II and increases rapidly in spring (Makkonen et al., 2014). The low nearby ammonia availability suggests that the wintertime nitrate is long–range transport.

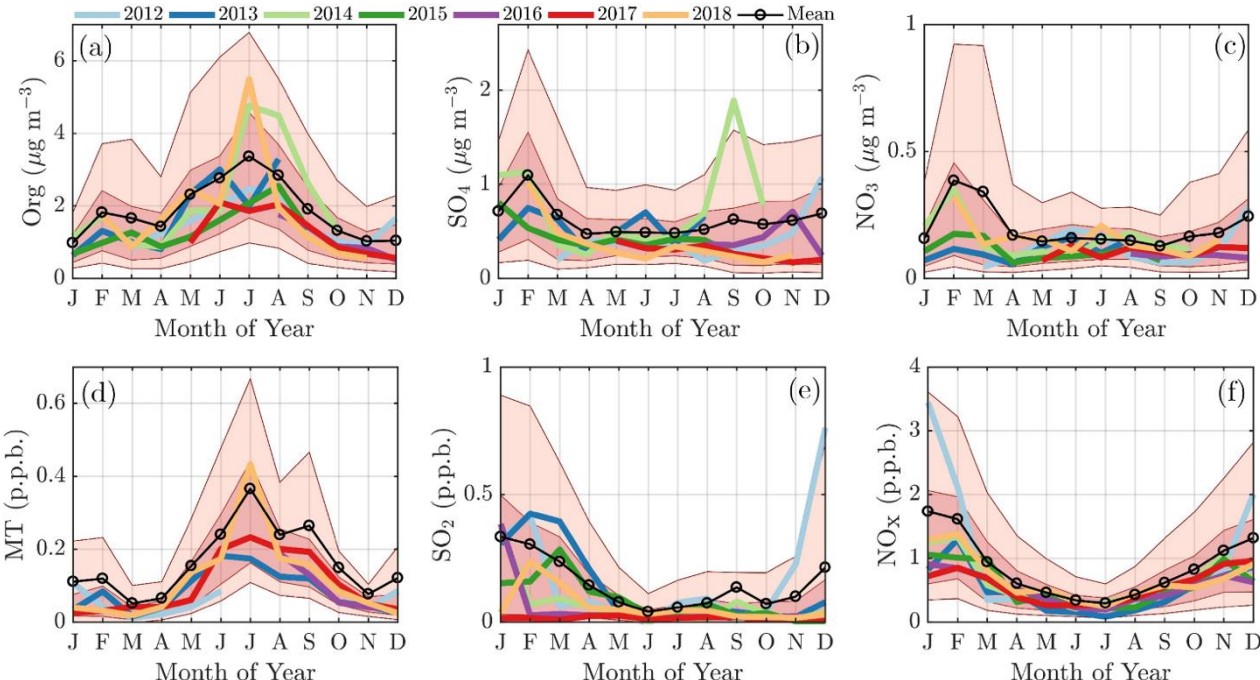

484

**Figure 4** The monthly cycles of organics (panel a), sulphate (panel b) and nitrate (panel c) concentrations in the NR–PM$_1$, and their major precursors, i.e. monoterpenes (panel d), sulphur dioxide (panel e), and nitrogen oxides (panel f). The median monthly concentrations for each year are individually displayed with the coloured solid lines. The black circled line represents the overall monthly mean values. The dark red shaded area is drawn between the overall 25$^{th}$ and 75$^{th}$ percentiles and the lighter red shaded area between the overall 10$^{th}$ and 90$^{th}$ percentile. The x–axes represent the time of the year and the y–axes in panels a–c mass concentration in µg m$^{-3}$, and d–f in parts per billion (p.p.b.).

Monoterpenes also show a non–zero loading during all winter months (Figure 4d). Their source is likely anthropogenic, such as the nearby sawmills, rather than biogenic due to the low ambient temperature limiting their biogenic emissions (Guenther et al., 1993;Hakola et al., 2012;Kontkanen et al., 2016). Their wintertime presence could be linked to their overall decreased photochemical sink and them being emitted to the shallower atmospheric boundary layer with little vertical dilution. The monoterpene mixing ratio starts increasing rapidly in April achieving its maximum monthly median value during July (Figure 4d), simultaneously with NR–PM$_1$ organics (Figure 4a). Previous studies have shown that monoterpene emissions increase exponentially with ambient temperature (Guenther et al., 1993;Aalto et al., 2015). As the current study does not incorporate monoterpene emission data, we visualise the behaviour of monoterpene mixing ratio with increasing ambient temperature in Figure 5b. The increase in monoterpene mixing ratio is likely a combined result from increased biological plant activity in the forest as well as the increased emissions due to higher temperature. Organic aerosol concentration behaves in a similar manner as a function of temperature, as depicted in Figure 5a. Such behaviour in organic aerosol loading is often attributed to biogenic SOA formation from BVOCs (Daellenbach et al., 2017;Stefenelli et al., 2019;Vlachou et al., 2018;Paasonen et al., 2013).

Besides biogenic SOA, also open fire biomass burning organic aerosol (BBOA) is a major contributor to summertime organics worldwide, from wild fires during dry conditions (Bond et al., 2004;De Gouw and Jimenez, 2009;Mikhailov et al., 2017;Corrigan et al., 2013). An enhancement in eBC concentration, a tracer for BBOA, was observed as a function of temperature at SMEAR II implying the possible presence of BBOA in the summertime sub–micron aerosol (Figure

5d). However, uncertainties can be attributed to the eBC concentration as scattering coatings, such as salts or even
photochemically aged SOA can also generate a lensing effect leading to an overestimation of eBC (Bond and Bergstrom,
2006;Zhang et al., 2018). Such effect could lead to a substantial overestimation of eBC especially in summertime, when
the organic loading is highest. Some certainty of the BBOA and BC enhancement with high temperatures can however
be retrieved from Figure 5c, where carbon monoxide (CO) mixing ratio also increases at relatively high ambient
temperatures (T > 15 °C). CO is known to be emitted from incomplete combustion processes. Nonetheless, as the increase
in eBC visualised in Figure 5d is less drastic than for monoterpenes, we suggest the biogenic SOA production as the major
organic aerosol source in summertime. While the quantification and separation of BBOA from SOA will be the topic of
an upcoming independent publication centred on the analysis of organic aerosol mass spectral fingerprints at SMEAR II,
we briefly introduce the behaviour of $f_{60}$. $f_{60}$, which equals the contribution of m/Q 60 Th signal to the total organic signal
recorded by the ACSM, is a marker for levoglucosan-like species originating from cellulose pyrolysis in biomass burning
(Schneider et al., 2006;Alfarra et al., 2007). $f_{60}$ is present at high percentages in fresh BBOA plumes (Cubison et al.,
2011), but decays due to BBOA photochemical aging into oxidised organic aerosol. The fairly rapid (in order of several
hours to days) photochemical aging of BBOA has been a topic of earlier chamber (Grieshop et al., 2009;Jimenez et al.,
2009) and ambient studies (DeCarlo et al., 2010;Cubison et al., 2011). Here, unlike CO and eBC, $f_{60}$ does not increase as
a function of temperature in the highest temperature bins, but stays rather constant albeit lower than the $f_{60}$ values recorded
under cold temperatures at SMEAR II (Figure 5e). As the possible wild fires contributing to SMEAR II CO and eBC
under high ambient temperatures also occur further away, it is likely that the BBOA is oxidised before detected at SMEAR
II. CO and eBC can be considered as more inert BBOA markers compared to $f_{60}$. The wintertime $f_{60}$ is likely linked to
wintertime biomass burning (for domestic heating purposes) emissions trapped in the shallow mixing layer. These
emissions are discussed more later on in the manuscript (see chapter 3.2 Diurnal variation of NR–PM$_1$ composition)."

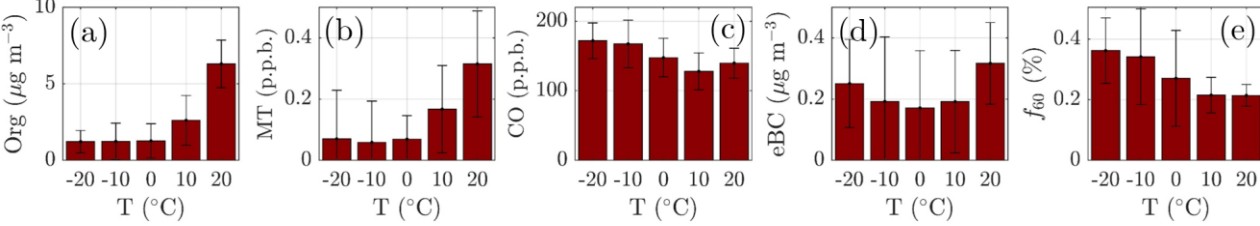


**Figure 5** The daily mean organic aerosol (Org, panel a), monoterpene (MT, panel b), carbon monoxide (CO, panel c), equivalent
black carbon (eBC, panel d) concentrations, and the fraction of the ACSM Org-signal made up by levoglucosan-like species ($f_{60}$, panel
e) recorded under different ambient temperatures. The values recorded assigned into 10 °C wide bins based on the daily ambient
temperature mean. The marker error bars show the standard deviation of the values in each bin.

### 3.1.1 Case study: The effect of warm summers on organic aerosol loading

The highest NR–PM$_1$ mass concentrations were detected during summers of 2014 and 2018. These summers were the
hottest during the whole measurement period (Figure 6a), and linked to persistent high pressure conditions (Sinclair et
al., 2019;FMI, 2014) . The non–parametric probability densities (Kernel distributions) for temperature for Julys 2012–
2018, displayed individually in Figure 4a, show clearly higher temperatures in July 2014 and 2018. Indeed, these months
were abnormally warm as July 2014 was 2.2°C, and July 2018 was 3.4°C higher than the 7–year July mean (16.6 °C).
Comparing to the 30–year July climate at SMEAR II (1981–2010) (Pirinen et al., 2012), the mean temperature in July
2014 was 2.8°C, and July 2018 4.0°C higher. As can be seen from Figures 4a&d, both organic aerosol and monoterpene
concentration positively responded to this temperature change with high median values. The same phenomenon is
visualised in Figures 4b&c through Kernel densities for particulate organics and monoterpenes, respectively, for July of
each year. The recorded organic aerosol concentration was 50% higher than the 7–year mean in July 2014, and 70%
higher in 2018. The monoterpene concentration in turn was 50% higher in 2018 compared to the mean of all available
July data (due to PTR–MS sensitivity issues, the 2014 data was chosen to be excluded from the analysis).

As the high–pressure weather ruling in Julys 2014 and 2018 further promoted clear–sky conditions, also the oxidation
capacity of the air was likely affected. This could have led to efficient monoterpene oxidation towards condensable low–
volatility products. Furthermore, their condensation onto particles could explain the observed high organic aerosol mass
concentration. The SOA formation enhancement as a function of temperature has also been investigated previously in a
modelling study, where a significant global increase in monoterpene–derived organic aerosol concentration was projected
to future, following different climate scenarios introduced by the Intergovernmental Panel of Climate Change (IPCC)
(Heald et al., 2008). Kourtchev et al. (2016) investigated the effect of high ambient temperature on biogenic SOA loading
and composition utilising measurement data from SMEAR II. They include summers 2011 and 2014 to their analysis,
where 2011 summer represents a significantly colder summer (average ambient temperature was 8°C less than in 2014).
By utilising ultra–high resolution off–line mass spectrometry on filter samples collected at SMEAR II, they detected a
significantly higher SOA oligomer content during 2014. Their results not only highlight the large increase in SOA mass
as a function of temperature, but also on the SOA composition differences affected by the large SOA content which
further influenced the CCN formation potential of SOA.

A quick revisit to the OCEC vs ACSM comparison performed earlier in the manuscript (Figure 2e) shows relatively high
Org/OC-values ($k = 1.68$) for July 2018, which further indicates of high oxygenation of OA (Aiken et al., 2008). However,
the time series of the ruling monthly Org/OC-values, visualised in Figure A.5, reveal an even higher Org/OC for August
2018. Further analysis of Org/OC recorded from SMEAR II is needed to answer whether such behaviour is frequently
occurring at SMEAR II, or whether July 2018 organic aerosol was less functionalised (oxidised) than usual due to higher
presence of primary organic aerosol, such as BBOA, than usual. Thus, not to link the organic aerosol increase to biogenic
SOA formation exclusively, we also investigated the presence of BBOA in the sub–micron aerosol during Julys 2014 and
2018 via Kernel distributions for eBC and $f_{60}$, presented in Figure 6d&e. The eBC distribution clearly hints towards
BBOA presence during July 2018, whereas July 2014 seems much less affected. Importantly, we also want to inform that
in July 2018 we had only one week of eBC data available.  The July 2018 eBC could be linked to the severe wild fires
occurring in Sweden during July and August 2018. The July eBC measurement period overlaps with the forest fire
occurrence period. Sweden also suffered from wild fires in August 2014, not depicted in Figure 6 that focuses on Julys
alone. The $f_{60}$ in turn follows the conclusions made earlier in the context of Figure 5 as the $f_{60}$ values remain low each
July, and approach the $f_{60}$ background levels of 0.3% (Cubison et al., 2011; Figure 6e). Importantly, such negligible $f_{60}$
signals were detected under the influence of an aged BBOA plume originating from Moscow and Northern Ukraine wild
fires at SMEAR II also in 2010 (Corrigan et al., 2013). An AMS, which was used as one of the measurement tools in the
campaign, detected mass spectra resembling oxidised organic aerosol during the biomass burning influence. These data
correlated well with multiple biomass burning markers including CO, potassium and acetonitrile despite the lack of
resemblance with fresh BBOA mass spectra with high $f_{60}$. Corrigan et al. (2013) finally attributed up to 25% of the organic
aerosol to BBOA originating from the Moscow and Northern Ukraine wild fires. 35% of the organic aerosol mass was
associated with biogenic SOA formation. The weather during the Corrigan et al. (2013) study period in 2010 was also
unusually warm ($T_\text{avg}$ = 20°C), and resembled summers 2014 and 2018 regarding the ruling anti-cyclonic circulation
pattern.

The frequency, duration and intensity of heatwaves are projected to increase in the future Finnish climate due to positive
pressure anomalies over Finland and to the east of Finland, as well as a negative pressure anomaly over Russia between
90 and 120°E (Kim et al., 2018). Moreover, the IPCC states that droughts and insect outbreaks, are projected to be boosted
in the warming climate (Barros et al., 2014). Recent findings by Zhao et al. (2017) show how such biotic and abiotic
stress factors enhance VOC emissions from plants that further contribute to organic aerosol after oxidation (Zhao et al.,
2017). Wildfires in turn are likewise likely to occur globally more frequently in the future due to increasing number of
long–lasting heatwaves (Spracklen et al., 2009). Indeed, BBOA loadings due to wildfires already show a slight increase
in the United States (Ridley et al., 2018). Based on these previous studies, as well as our observations together with the
Corrigan et al. (2013) study from SMEAR II, the increasing frequency of heat waves and wildfires will enhance the
particulate matter loading at SMEAR II in the future, and continuous long–term measurements, like the ones presented
here, will be important in monitoring such changes.

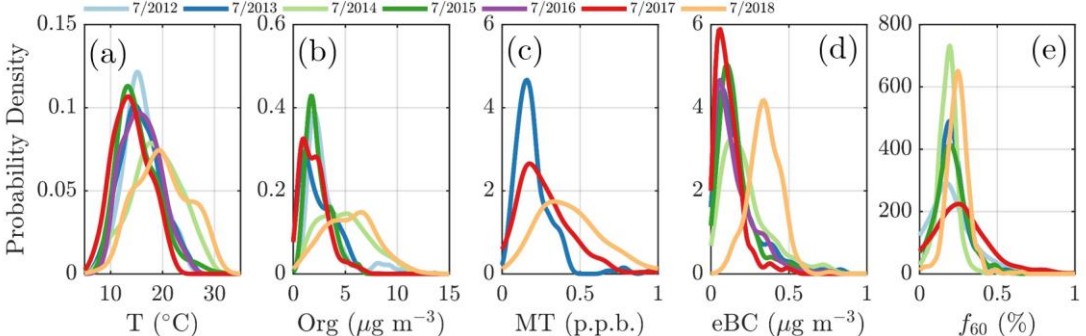

**Figure 6** Non–parametric probability densities (Kernel distributions) of temperature (panel a), organic aerosol (panel b), monoterpenes
(panel c), equivalent black carbon (panel d) concentrations, and the fraction of the ACSM organic signal made up by levoglucosan-
like species ($f_{60}$; panel e) during individual Julys across the measurement period (2012–2018). The data availability for eBC was one
week in July 2018. The x–axes represent the T, Org, MT and eBC values recorded, respectively and the y–axes the non–parametric
probability densities. Briefly, the Kernel distributions are similar to smoothed histograms of the measurement data. This visualisation
was chosen to avoid assumptions of the nature of distribution that might hide important features of the measurement data if presented
with normal distributions, for instance.

**3.1.2 Case study: Sulphate transport from Holuhraun flood lava eruption**
The sulphate loading in September 2014 represents the largest outlier in Figure 4b with the average mass concentration
five times greater than the overall September mean. The mean $SO_2$ concentration during September 2014 was 0.66 p.p.b.,
which is 0.50 p.p.b. higher than the mean $SO_2$ mixing ratio representing all the Septembers during the measurement period
(0.16 p.p.b.). The elevated median $SO_2$ concentration during September 2014 is also visible in Figure 4e. To investigate
the source of sulphur, we displayed the full atmospheric column $SO_2$ concentration during September 2014 utilising
satellite observations (Figure A.6a). The $SO_2$ concentration hot spot was located in Iceland and is linked to the fissure
Bárðarbunga–Veiðivötn eruption at Holuhraun (Aug 31st 2014 – Feb 28th 2015) that yielded 20–120 kilotons a day of $SO_2$
(Schmidt et al., 2015). The concentration above SMEAR II was obviously not comparable to the loading near the eruption
site. However, based on a rough trajectory analysis (Figure A.6b) we link our observations of elevated sulphate and $SO_2$
during September 2014 to the diluted plume from Holuhraun.
**3.2 Diurnal variation of NR–PM$_1$ composition**
The year–to–year variation in the NR–PM$_1$ monthly median seasonal cycles shows rather consistent behaviour throughout
the measurement period and even the overall 10[th] percentile of the PM–data suits the bimodal trend discussed in the
section above. The 10[th] percentile also agrees with the seasonal trends associated with individual NR–PM$_1$ chemical
species, i.e. organics, sulphate and nitrate as well as their precursors (Figure 4). Few outliers observed are discussed in
the chapters above (see chapters 3.1.1 Case study: The effect of warm summers on organic aerosol loading and chapter
3.1.2 Sulphate transport from Holuhraun flood lava eruption). As the year–to–year variability between different years is
rather minimal, we decided to investigate the overall median temporal behaviour of aerosol chemical composition further
via Figure 7. The subplots in this figure are based on data matrices of median diurnal cycles (1h resolution) for every two
weeks of a year ($24 \times 26$ matrix). The matrices are visualised with contour plots (*contourf*, MATLAB 2017a) except for
Figure 7h, due to the high noise level of the time trace.

Neither the NR–PM$_1$ concentration nor its chemical species have large diurnal variability during wintertime (Figure 7a)
due to low solar radiation and the lack of diurnal variability in ambient temperature (Figures 1a&b) prolonging the life
time of aerosols. Thus, the wintertime chemical composition of NR–PM$_1$ stays stable over the course of the day (Figures
7b&d). As wintertime PM is presumably mostly long–range transport, its components' diurnal patterns are less obvious
due their cumulative build–up in the atmosphere. For example, as sulphate aerosols, the most prominent inorganic species,
are long lived due to their low volatility, we do not expect sulphate to have diurnal variation in wintertime because of the
lack of major $SO_2$ sources at SMEAR II's proximity. The ammonium mass concentration lacks diurnal pattern as well
and peaks at the same time of the year as sulphate. The degree of aerosol neutralisation by ammonia can be estimated by
the ratio between the measured ammonium and the amount of ammonium needed to neutralise the anions  detected by the
ACSM (termed "NH$_4$ predicted") (Zhang et al., 2007b). The overall ratio was 0.66 hinting towards moderately acidic
ammonium sulphate aerosols (Figure A.7&Figure 7h), though the uncertainty in this value is high due to the low loadings
of ammonium at SMEAR II. We also acknowledge that the ratio between measured and predicted ammonium
concentration is not fully accurate for acidity estimations, and if such are needed, a better estimation could be provided
with thermodynamic models. The temporal variation of the ammonium balance does not show diurnal variability either,
but a very modest decrease during January (Figure 7h), when the ambient temperature was the lowest. In the case of
wintertime organic aerosol, the lack of a diurnal trend (Figure 7c) indicates that nearby residential heating (expected
mainly in evenings) emissions are not a dominating source of organics at the site despite their clear presence in the
wintertime aerosol as depicted by the seasonal $f_{60}$ trend depicted in Figure A.4. In general, the lack of a distinct diurnal
pattern rather hints towards long–range transported organics. Such conclusions can also be made based on the rather high
Org/OC linear regression slope for February data ($k = 1.65$) depicted in Figure 2e (see also Figure A.5). As mentioned
earlier, a high Org/OC-ratio indicates a higher degree of functionalisation/oxidation of organic aerosol (Aiken et al.,

656  2008).


From March onwards, when the solar radiation flux has significantly increased, the aerosol chemical composition starts
to show modest diurnal variability. The ratio between organic and inorganic aerosol chemical species (OIR) exhibits

diurnal variability from March to October, when also ambient temperature has strong diurnal variation. The OIR achieves its minimum during daytime and maximum during night (Figure 7b). In other words, particles have the highest organic fraction during night-time and early mornings. The organic aerosol mass concentration increases during night (Figure 7c), likely due to more efficient partitioning of semi–volatile species into the aerosol phase. This effect is seen even more clearly in the nitrate concentration (Figure 7e), with a strong diurnal pattern largely tracking the diurnal temperature trends over the year.

The nature of particulate nitrate can be estimated via fragmentation ratios of $NO^+$ and $NO_2^+$ ions detected by the ACSM as described by (Farmer et al., 2010) for the AMS. A higher ratio (> 5) generally means a greater presence of organic nitrates and a lower ratio (2–3) indicates inorganic ammonium nitrate. As the ACSM has a low mass resolving power, we here estimate the ratio between $m/Q$ 30 and $m/Q$ 46 Th as a proxy for the $NO^+ : NO_2^+$ –ratio. We note that there is possible interference of organic mass fragments at these $m/Q$–ratios. Nonetheless, we observe that the wintertime nitrate resembles ammonium nitrate and the summertime nitrate hints towards the presence of organic nitrates (Figure 7g). This is in line with the recent study stating that more than 50% of the nitrates detected in the sub–micron particles at SMEAR II are estimated to contain organic nitrate functionalities (Äijälä et al., 2019). However, we should stress the fact that the data coverage of wintertime was limited in the Äijälä et al. (2019) study that could lead to an overestimation of annual organic nitrate mass fraction. Such high organic nitrate fraction could also explain some of the scatter observed in the ACSM $NO_3$ and MARGA $NO_3$ comparison discussed earlier in the manuscript (Figure A.3 and chapter 2.5 ACSM chemical speciation validation). We observe no clear diurnal pattern in the fragmentation ratio.

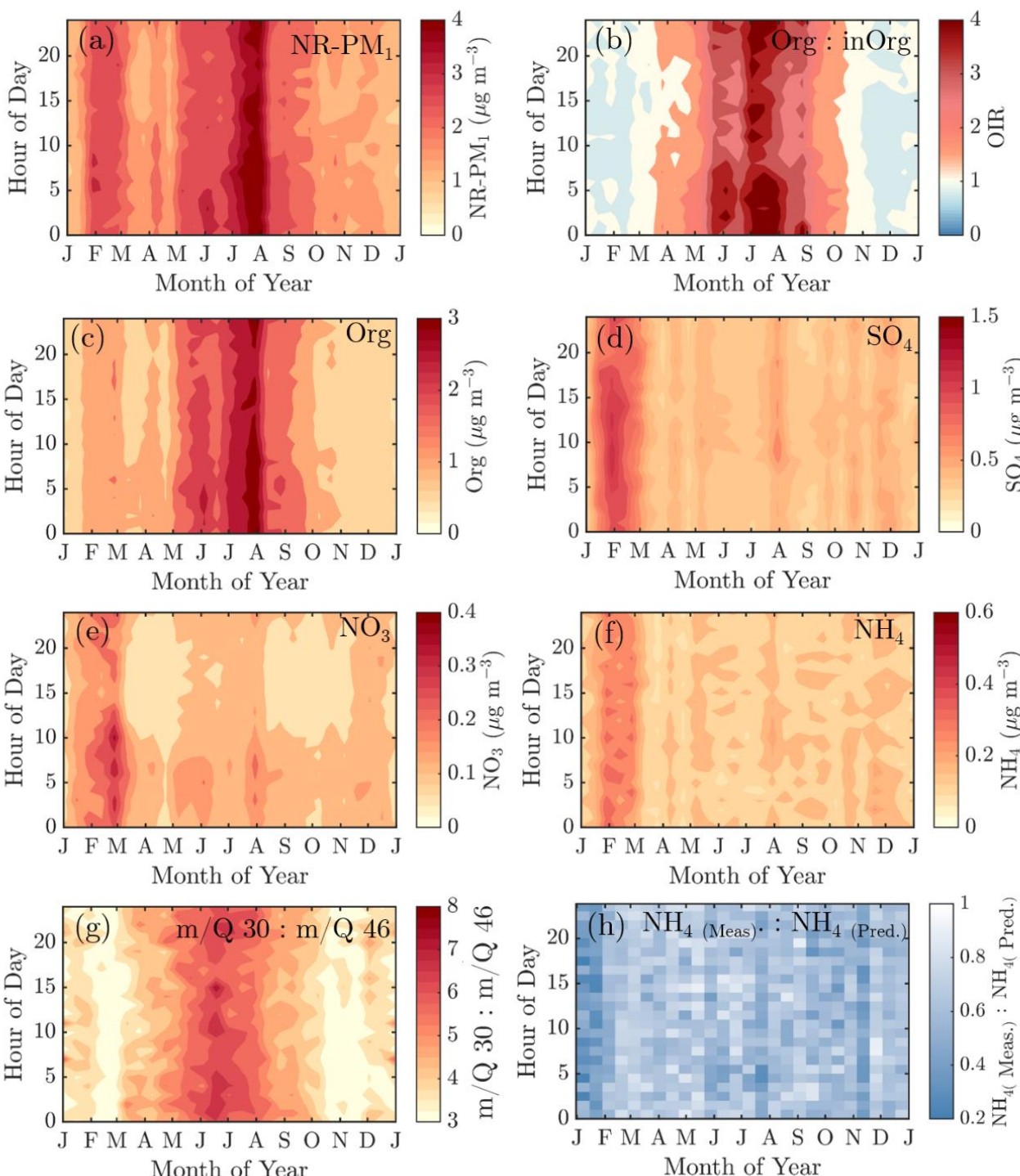

681

**Figure 7** The median diurnal cycles of NR–PM₁ (panel a), Organic–to–inorganic ratio (panel b), organic aerosol (panel c), sulphate (panel d), nitrate (panel e), ammonium (panel f), *m/Q* 30 : *m/Q* 46 Th fragmentation ratio (panel g), and the ratio between measured and predicted ammonium (panel h). The y–axes represent the local time of day (UTC+2) and x–axes the month. The color scales present the mass concentration (panels a, c–f) or ratios (panels b, g–h). Note that the scaling of the color bar is different in all of the figures. Panels a-g include interpolation of the 14 d × 1 h resolution data grid based on the MATLAB 2017a *contourf* function. Panel h has no interpolation involved due to the high noise level of the variable. The plot is produced with MATLAB 2017a *pcolor* function.

688

### 3.3 Wind direction dependence

The wind direction plays a key role together with other meteorological conditions determining the aerosol chemical composition at SMEAR II. While the sections above focus more on the role of radiation and temperature on sub–micron aerosol composition, this section explains the role of wind direction and speed. We want to stress that this section does not include any definite geographical source analysis of the NR–PM$_1$ components. A detailed trajectory analysis is a better tool for understanding the actual footprint areas of air pollutants as wind direction analysis might lead to a systematic bias in the pollutant origins due to prevailing weather patterns.

### 3.3.1 Wind sector dependent diurnal cycles of organics and sulphate

To explore the wind direction dependence of the seasonal cycles of the main NR–PM$_1$ chemical species, organics and sulphate, we visualised their monthly median diurnal cycles with 4–hour time resolution ($12 \times 6$ matrix) for four different wind direction bins: 0–90° (I), 90–180° (II), 180–270° (III), and 270–360° (IV) in Figure 8. The frequency of different wind directions are depicted in Figure A.1, showing that e.g. sector I was the least likely, while wind from sector III was the dominant direction.

The highest organic aerosol loading was observed during summer for all of the wind direction bins (I – IV) with rather modest diurnal variability, perhaps due to the coarse time resolution used (Figure 8, left panels). The greatest organic aerosol concentration was associated with sector II that covers the direction of the Korkeakoski sawmills located 6 – 7 km to the SW (Figure 8c). Moreover, the February peak in organic aerosol was also most distinguishable from sector II (Figure 8c). Sulphate aerosol in turn was mostly detected with winds from sector I and II (Figure 8, right panels). Sector I shows a general wintertime enhancement (Figure 8b), whereas sector II shows a clear maximum during February (Figure 8d). The westerly sectors (III&IV) were associated with cleaner air (Figures 8e–h).

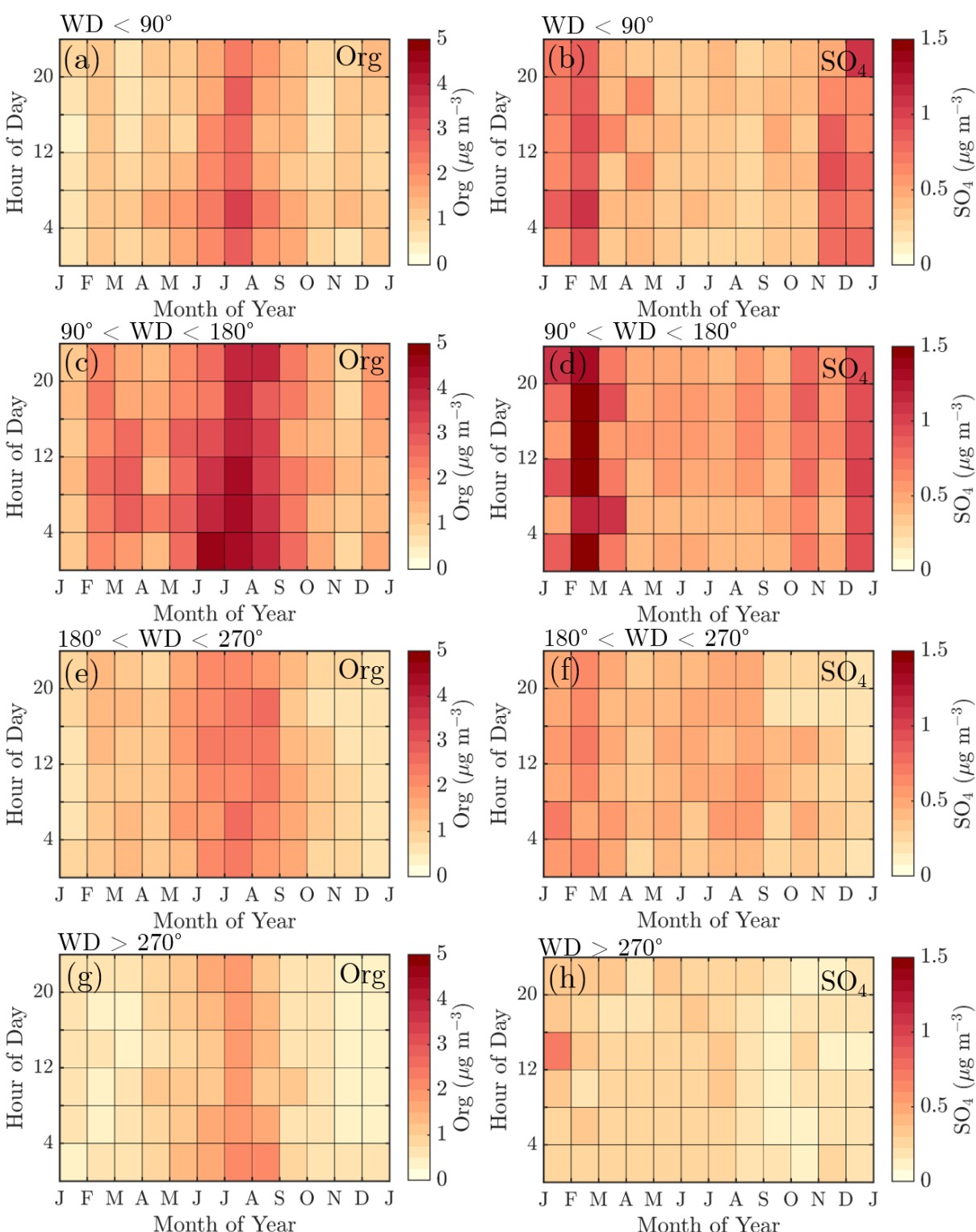

711

**Figure 8** Diurnal cycles of organic aerosol and sulphate divided into different wind direction bins: Panels a–b: wind direction <90°, panels c–d: wind direction 90–180°, panels e–f: wind direction 180–270°, and panels g–h: wind direction >270°. The y–axes represent the local time of day (UTC+2), and the x–axes the time of the year. The color scales represent the organic aerosol and sulphate aerosol mass concentrations in µg m$^{-3}$. Figure A.1 introduces the likelihoods of each wind direction bin via a traditional wind rose plot.



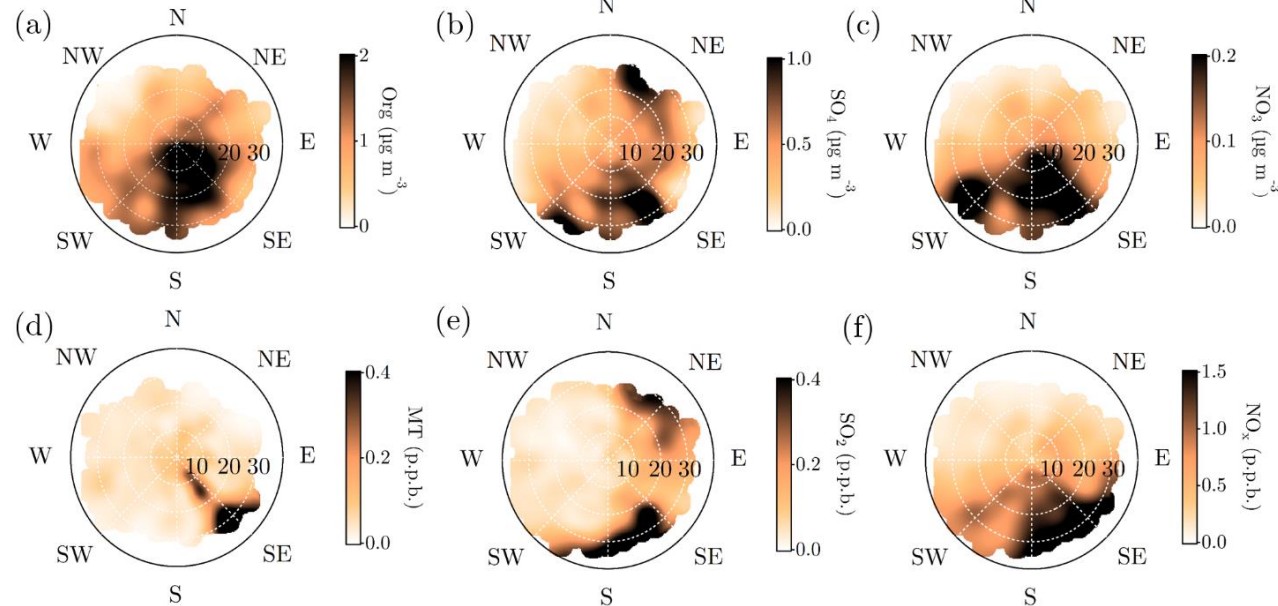


**Figure 9** Openair polar plots for organic aerosol (panel a), sulphate (panel b), nitrate (panel c), monoterpenes (panel d), SO₂ (panel
e), and NOₓ (panel f). The distances from the origin indicates wind speeds in km h⁻¹. The wind speed grid lines are presented with
white dashed circles. The colour scales represent the concentrations observed with each wind speed and direction combinations. As the
figures do not indicate any likelihood of the wind speed and distance combinations, Figure A.1 is important to keep in mind while
interpreting them. Briefly, N–NE–E is the least probable wind direction, whereas S–SW–W is the most likely. Wind speeds generally
stay below 20 km h⁻¹.

**3.3.2 Openair: Organics and monoterpenes**
Finally, we investigate the aerosol chemical composition dependence of wind speed and direction utilizing openair polar
plots. As the polar plots do not take into account the frequency of certain wind direction and speed combinations, Figures
1c&d and Figure A.1 are important when drawing conclusions based on them.

Organic aerosol concentration at SMEAR II increased with S–SE winds as already visualised also in Figure 8c (Figure
9a). The monoterpene mixing ratio also peaked, with a more narrow range of wind directions, analogous with the direction
of the nearby Korkeakoski sawmills (Figure 9d). With higher wind speeds, monoterpenes were also observed from a
wider span of wind directions. Organic aerosol showed wind speed dependence with S–SE winds with lower
concentrations associated with wind speed exceeding 25 km h⁻¹ (ca. 6.9 m s⁻¹). A possible explanation is that the
monoterpene emissions from the sawmills did not have time to oxidise and form SOA with such high wind speeds before
reaching SMEAR II. Organic aerosol concentration was relatively constant outside the sawmill interference, though the
lowest loadings were detected when air masses arrived with wind speeds exceeding 20 km h⁻¹ (ca. 5.5 m s⁻¹) from the
NW sector. In contrast, monoterpene mixing ratio was rather constant with varying wind directions and wind speeds,
obviously again apart from the sawmills direction (approximately 130°). Similar observations of the wind direction
dependence of monoterpene mixing ratios have been reported before, with a subsequent organic aerosol mass
concentration increase at SMEAR II with SW winds (Eerdekens et al., 2009;Liao et al., 2011).

A simplified seasonal analysis on aerosol chemical composition wind dependence was performed by investigating the
openair polar plots for all data recorded in February (**Figure A.1**Figure A.8) and July (Figure A.9). Korkeakoski sawmills
represented the main monoterpene source in February as the concentration coinciding with air masses arriving from other
directions was negligible (Figure A.8d). In February, the sawmill emissions did not significantly enhance the organic
aerosol concentration at the site, due to low oxidation rates (monoterpene life time up to 10 h; Peräkylä et al., 2014) and
higher wind speeds (Figure A.8a). The organic aerosol concentration approached zero with NW winds during February
regardless of the wind speed. A major wind speed influence can be observed with SW winds, as higher wind speeds
coincide with elevated organic loading.

In July, the monoterpene mixing ratio increased regardless of the wind direction due to increased biogenic emissions from
the surrounding forest, but also the sawmill influence remained elevated (Figure A.9d). The monoterpene life time in July
is roughly two hours (Peräkylä et al., 2014) indicating an efficient photochemical sink. Thus, monoterpene sources are
likely not that far. The organic aerosol concentration was clearly overall elevated, however the overall easterly
interference was more pronounced compared to February (Figures A.9a). It could be linked to the high pressure systems
often associated with easterly winds that bring warm air and clear sky conditions to SMEAR II promoting BVOC
emissions and SOA formation as discussed earlier in the paper.

**3.3.3 Openair: Sulphate and SO$_2$**
Relatively high concentrations of sulphate aerosols and sulphur dioxide were detected with N–NE and SE–SW winds
(Figures 9b&e). Riuttanen et al. (2013) performed a HYSPLIT trajectory analysis for SMEAR II for 1996–2008 with SO$_2$
concentration fields showing similar results. They attribute the detected SO$_2$ to anthropogenic emission sources in St.
Petersburg, Baltic region, Kola Peninsula and the SE corner of the White Sea. In addition to the major emission sources
introduced by Riuttanen et al. (2013), also paper and pulp industry are major known SO$_2$ emitters. Several paper and pulp
mills are situated in Finland, mostly NE and SE from SMEAR II (Metsäteollisuus, 2018). Another national major SO$_2$
source is certainly the Kilpilahti (Porvoo) oil refinery, located ~200 km S-SE from SMEAR II. This area represents the
most extensive oil refinery and chemical industry in the Nordic countries, and the SO$_2$ concentrations measured downwind
from the area have been close to those obtained from Kola Peninsula outflow (Sarnela et al., 2015).

Large emission sources located SW of SMEAR II listed by EMEP (European Monitoring and Evaluation Programme)
did not stand out in the analysis performed by Riuttanen et al. (2013), but a wind direction dependence visible in the
current study, associated only with high wind speeds. Similar wind speed dependence was observed with SE–S and N–
NE winds as the concentration of sulphate and SO$_2$ clearly increased when wind speeds exceeded 20 km h$^{-1}$ (ca. 5.5 m s$^{-}$
$^1$). Such wind speed dependence can be observed with long–range transported air pollutants: their transport is generally
more efficient with higher wind speeds. The results presented here are also consistent with hygroscopicity measurements
conducted at SMEAR II (Petäjä et al., 2005), where the hygroscopic growth factor was greatest when SO$_2$ rich air arrived
fast to the station from the NE.

NE and SE represent the major SO$_2$ sources in February. The NE SO$_2$ was detected with lower wind speed dependence
than generally observed (Figures A.8b&e). The lifetime of SO$_2$ is dependent on wet and dry deposition, and oxidation to
sulphate (photochemistry or aqueous phase chemistry in cloud droplets). These factors influence the likelihood of
detecting $SO_2$ from distant sources. The higher wintertime concentrations are also linked to the atmospheric boundary
layer dynamics, as discussed earlier. The SW and SE–S winds with wind speeds exceeding 16 km h$^{-1}$ (ca. 4.4 m s$^{-1}$) were
associated with sulphate during February (Figure A.8b). Sulphate was detected also with a wide range of wind directions
during low wind speeds. In the case of low wind speeds, it is hard to determine the wind direction accurately. However,
it was clear that sulphate was not associated with W–NW winds, as shown previously in the paper (Figure 8, right panels).

The sulphate openair polar plots for July (Figure A.9b) reveals that the sulphate transport was more wind speed dependent
than in February. Moreover, the wind directions linked to sulphate presence at SMEAR II were NW–N, NE, and E–SE,
but observed only when the wind speeds exceeded 16 km h$^{-1}$ (ca. 4.4 m s$^{-1}$).  $SO_2$ was only observed with wind speeds
exceeded 16 km h$^{-1}$ (ca. 4.4 m s$^{-1}$) with NE winds (Figure A.9e). High wind speeds are needed in July to transport rather
short–lived pollutants, such as $SO_2$, to SMEAR II from distant sources.

As can be seen from Figures 9e, A.8e & A.9e, elevated $SO_2$ concentrations (dark areas in the concentration fields) are
associated with very specific, rather narrow ranges of easterly (mainly NE and SE) wind directions, and elevated wind
speeds ($> 16 - 20$ km h$^{-1}$). These figures illustrate the sensitivity of the recorded $SO_2$ concentration towards even moderate
wind direction and speed variations. As wind direction and speed can vary significantly in an inter-annual scale, also
inter-annual variability in $SO_2$ concentration can be expected. This could finally explain the $SO_2$ inter-annual variability
highlighted especially in winter months (Figure 4e). Such strong variability is not reflected further in the sulphate aerosol
(Figure 4b) year-to-year scales due to its long lifetime and build-up in the atmosphere.

**3.3.4 Openair: Nitrate and NO$_x$**
The nitrate concentration field visualised in Figure 9c was highest when wind blew from SE– SW. No wind speed
dependence could be attributed to the nitrate from E–SE, whereas for SW, nitrate concentration clearly elevated when
wind speed exceeded 20 km h$^{-1}$ (ca. 5.5 m s$^{-1}$). NO$_x$ concentration, in turn, was not significantly elevated with SW winds
regardless of the wind speed, but shows similar behaviour to nitrate with SE–S winds (Figure 9f). The nitrates arriving
with SW likely spend more time in the atmosphere than in the case of SE–S source. A previous study focusing on organic
nitrates at SMEAR II linked their occurrence to SE winds (Kortelainen et al., 2017). They suggest night–time nitrate
radical oxidation of sawmill BVOCs as their major source. The same study attributes inorganic ammonium nitrate with
SW winds. The study was conducted in spring–time. Also our results suggested an increased organic nitrate presence in
spring compared to wintertime (Figure 7g).

In February, the nitrate concentration field resembles the overall concentration field depicted in Figure 9c, but highest
loadings were typically associated with low wind speeds from S–SE (Figure A.8c). The reason for not observing nitrate
with high wind speeds could be the fact that there is not enough time for nitrate aerosol formation. NO$_x$ concentration
was overall elevated between NE and SE, and the clean SE–N sector had negligible NO$_x$ loading (Figure A.8f). Despite
the NO$_x$ availability in the North, no nitrate aerosol was observed. This could be due to limited ammonia availability in
winter time. Most NO$_x$ was detected with E–SE winds when wind speed was 8–16 km h$^{-1}$ (ca. 2.2–4.4 m s$^{-1}$).

In July, SW winds blew most of the nitrate to SMEAR II (Figure A.9c). However, also slightly elevated concentrations
can be observed with S–SE winds (Figure A.9c).  The nitrate associated with SW winds again requires high wind speeds.
The NO$_x$ concentration was significantly lower in July compared to February, as already shown in Figure 4f (Figure A.9f).
No clear wind speed dependence was observed.

**3.3.4 Openair: Ammonium and ion balance**

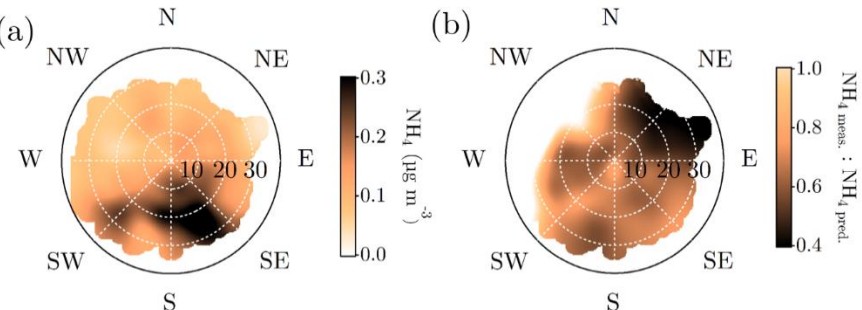


**Figure 10** Openair polar plots for ammonium (panel a), and the ratio between measured and predicted ammonium (panel b). The
distances from the origin indicates wind speeds in km h$^{-1}$. The wind speed grid lines are presented with white dashed circles. The color
scales represent the concentration (panel a) and the unitless ammonium ion balance ratio (panel b) observed with each wind speed and
direction combinations. As the figures do not indicate any likelihood of the wind speed and distance combinations, Figure A.1 is
important to keep in mind while interpreting them. Briefly, N–NE–E is the least probable wind direction, whereas S–SW–W is the
most likely. Wind speeds generally stay below 20 km h$^{-1}$.

The overall polar plot for ammonium, visualised in Figure 10a, did not show elevated abundance with N-NE winds in
contrary to sulphate polar plot (Figure 9b). Moreover, the ammonium ion balance showed lowest values with N–NE winds
that often carry the sulphate–rich aerosols to SMEAR II (Figures 9b&10b). Such observations hint towards acidic
aerosols. Riva et al. (2019) observed acidic aerosols likely originating from the Kola Peninsula that support this
hypothesis. Moreover, the particle acidity further drove chemical transformations in the aerosol organic leading to a higher
presence of oligomers in the aerosol. Also the hygroscopicity analysis carried out at SMEAR II back in 2005 showed how
the particles arriving from NE were most hygroscopic  (Petäjä et al., 2005) that is a property boosted in acidic aerosols.
The clean NW sector shows bright values for the ammonium balance field. Here, the ammonium balance exceeds one
due to the noisiness of the data introduced by both ammonium and nitrate used in the ammonium balance calculation
being below their detection limits during NW winds.
**4 Conclusions**
To better understand the boreal forest aerosol, an aerosol chemical speciation monitor (ACSM) was installed for long–
term monitoring of sub-micron aerosol chemical composition in 2012 at research site of SMEAR II. The measurements
continue to this day. Such measurements at the site had been previously conducted only in short–term intensive
measurement campaigns, leaving our understanding of the seasonal and year–to–year variability lacking. The current
study spans over the first seven years (2012–2018) of on–line monitoring of the sub–micron non–refractory aerosol
composition, finally providing this missing piece in SMEAR II aerosol documentation.

The median mass concentration over the measurement period was 2.3 µg m$^{-3}$ (1.2 and 4.0 µg m$^{-3}$ for the 25[th] and 75[th]
percentiles, respectively) of which 68% was organics, 20% sulphate, 6% nitrate, and 6% ammonium. Chloride
concentrations in the non–refractory sub–micron particles were negligible (< 1%).  As many factors, such as ambient

temperature, solar radiation, atmospheric boundary layer height and wind influence the aerosol particle concentrations and trace gas emissions, oxidation and volatility, we observed a clear seasonal cycle in NR-PM$_1$ loading and composition.

During warm months, biogenic VOC emissions increase, and upon oxidation, produce SOA which represents a major source of PM at SMEAR II. Organic aerosol mass concentration achieved its annual maximum in July (3.3, 1.7, and 4.6 µg m$^{-3}$ for median, 25$^{th}$ and 75$^{th}$ percentiles, respectively) that further lead to the annual maximum in the total NR-PM$_1$ loading (4.2, 2.2, and 5.7 µg m$^{-3}$ for median, 25$^{th}$, 75$^{th}$ percentiles, respectively). Organics on average made up 80% of the NR-PM$_1$ in summer. During the exceptionally hot Julys of 2014 and 2018, the organic aerosol concentrations were up to 70% higher than the 7–year July mean. Most of the mass could be associated with increased biogenic SOA production. The projected increase of heat wave frequency over Finland (and in general) will most likely influence the loading and chemical composition of aerosol particles, and subsequently affect the Earth's radiative balance. Also from this perspective, continuing the long–term measurements at SMEAR II is essential.

Winter months indicate low amounts of solar radiation and a shallow boundary layer. NO$_x$ and SO$_2$, the main precursors for particulate nitrate and sulphate, respectively, achieved their maximum mixing ratios during the darkest months while emitted into the shallow boundary layer during the period of low photochemical activity. These species are generally emitted in combustion processes that lead to high wintertime concentrations both due to the additional need of residential heating as well as the shallow boundary layer prohibiting their vertical mixing. The maximum wintertime NR-PM$_1$ concentration was most commonly detected in February, and explained by an enhancement of inorganic aerosol species. The particulate sulphate and nitrate peaked in February, which was later than their precursors, as a combined result of wind patterns, deposition mechanisms and photochemistry affecting their formation and removal rates. The contribution of inorganic aerosol species was ca. 50% of the total NR-PM$_1$ (2.7, 1.6, 5.1 µg m$^{-3}$ for median, 25$^{th}$, 75$^{th}$ percentiles, respectively) in February of which 30% was sulphate, 10% nitrate and 10% ammonium. Importantly, much of these inorganic aerosol species were most likely from long-range transport. If emission regulations regarding SO$_2$ and NO$_x$ become stricter in the future in Europe, and especially in Russia, the wintertime NR-PM$_1$ might decrease significantly at SMEAR II.

To our understanding, this is the longest time series reported describing the aerosol chemical composition measured on–line in the boreal region. Long-term monitoring of changes introduced by emission regulations together with the changes introduced by the changing climate, are crucial for understanding the aerosol-sensitivity of the (boreal) climate. Thus, we keep the ACSM measurements on going at SMEAR II to obtain an even longer data set. The data presented here will be publicly available, and we welcome collaborative work in utilising this information for broadening the understanding of the boreal environment.

**Data availability**

The ACSM data are available at EBAS database (http://ebas.nilu.no/). The trace gas and meteorology data are available at the SMART SMEAR data repository (https://avaa.tdata.fi/web/smart). Other data are available upon request from the corresponding authors.

**Author contributions**
LH, MÄ, ME, TP, MK, and DW designed the study. LH, MÄ and MA performed the ACSM measurements. LH processed and analysed
the ACSM data. JA and PR performed the PTR-MS measurements and data processing. HK provided and processed the Dekati impactor
data. PA performed the DMPS measurements and data processing. UM provided and processed the MARGA data. KL provided and
processed the Aethalometer data. DA performed satellite and trajectory analysis in Figure A.6. LH performed the overall analysis, data
visualisation and wrote the paper. ME supervised the process. All authors commented and edited the paper.
**Competing interests**
The authors declare no conflict of interest.
**Acknowledgements**
First, we thank SMEAR II staff, Petri Keronen, Erkki Siivola and Frans Korhonen for measurement maintenance and support. We
thank the ACMCC, COST–COLOSSAL, and Aerodyne Research for the guidance towards high–quality instrument operation. We
thank Otso Peräkylä and Jenni Kontkanen for useful discussions. For financial support, we acknowledge the European Research
Council Starting Grant COALA. We thank the anonymous reviewers for their expertise.

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

    **Appendix A: Supporting figures**

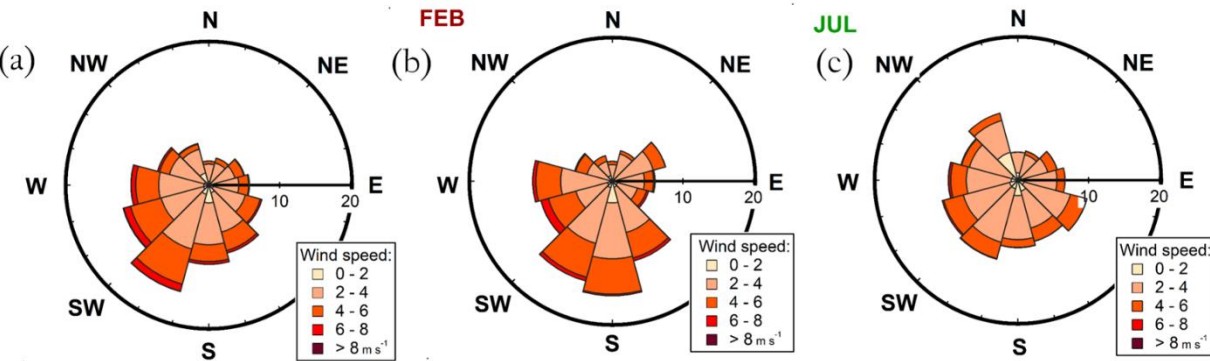

1227

**Figure A.1** Wind rose diagrams during the overall measurement period (panel a), February (panel b), and July (panel c). The distance from origin reflects the likelihood of each direction (%) and the color scale reflects the likelihoods of different wind speeds associated with the direction.

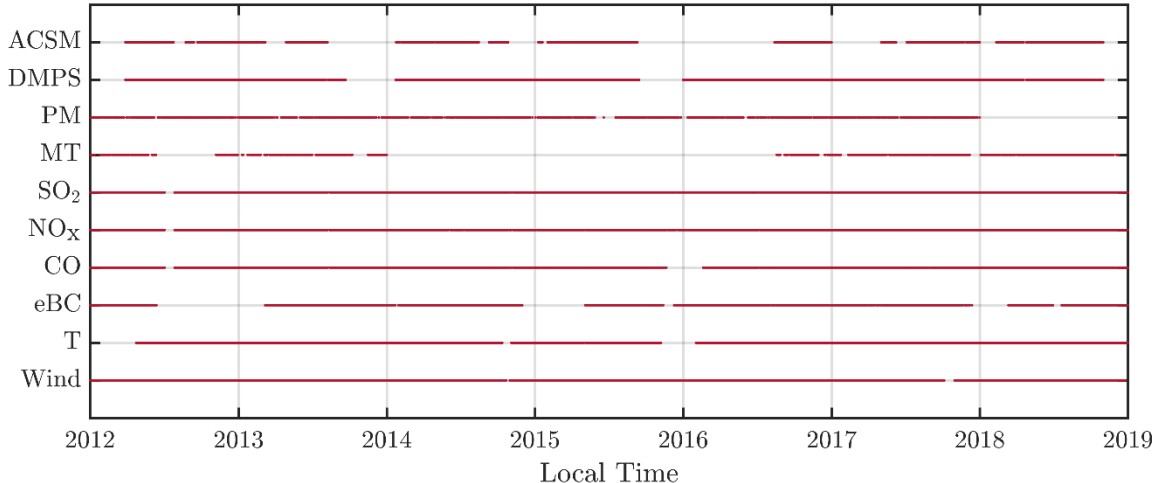

1231

**Figure A.2** Data availability during the measurement period. The instrument/measurement parameter is on the y–axis and time is on the x–axis. Gaps in the red line correspond to times when no data was available.

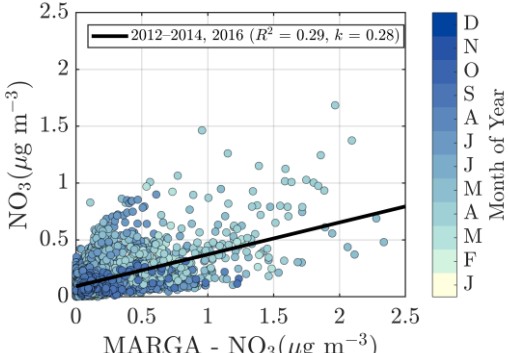

1234

**Figure A.3** The ACSM nitrate vs the PM$_{2.5}$ nitrate detected with MARGA–2S. The color coding represents the month of the year. The black line represents the overall linear fit.

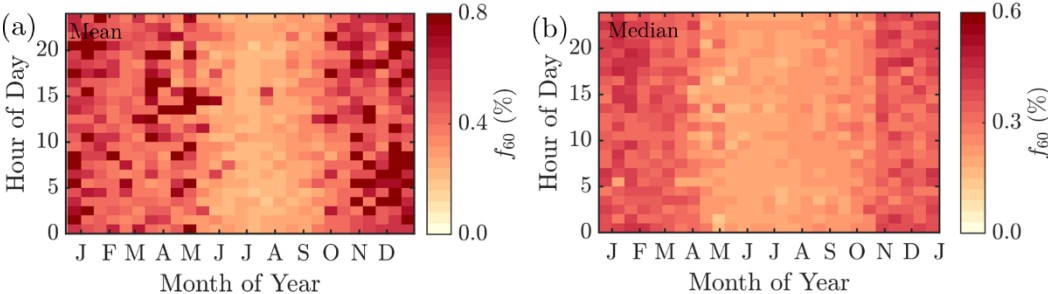

1237

**Figure A.4** The mean (a-panel) and median (b-panel) $f_{60}$ (the fraction of $m/Q$ 60 Th signal of the total OA signal) values derived from the ACSM measurements (2012–2018 at SMEAR II). The x-axes represent the time of the year and the y-axes the hour of the day (UTC+2). The coloured pixels represent the $f_{60}$ values. Note the different colour scales between the mean and median figures. It is also worth mentioning that due to the rather low signal to noise ratio of the ACSM, the $f_{60}$ estimates can be very noisy. To avoid the weight of the high and low noise extremes in the a-panel (mean $f_{60}$), only the range of $0 \geq f_{60} \leq 1$ were included in the $f_{60}$ mean field calculation.

1243

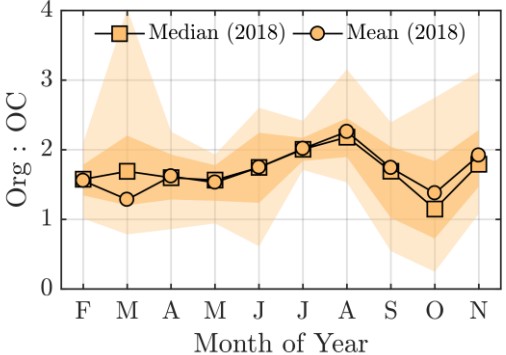

1244

**Figure A.5** Organic carbon concentration (OC) vs the organic aerosol concentration (Org) in 2018 at SMEAR II. The darker yellow shadings indicate the area between the 25th and 75th percentiles and the lighter yellow the area between the 10th and 90th percentiles.

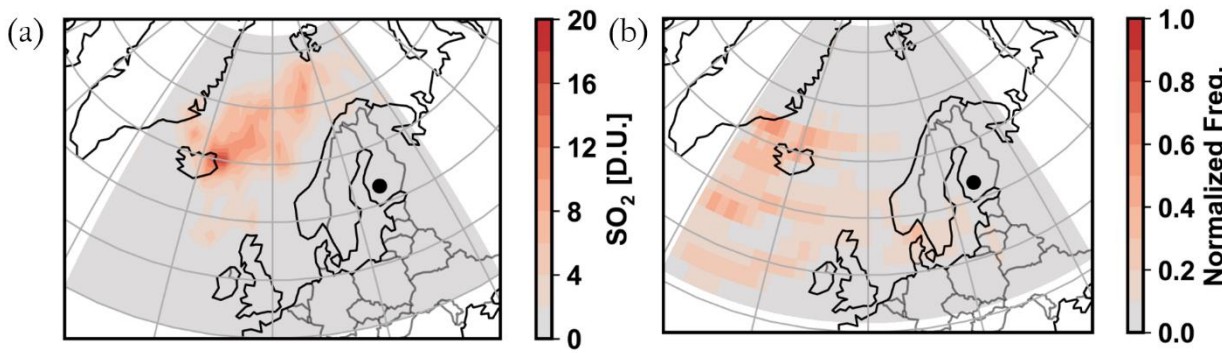

1247

**Figure A.6** (Panel a) Average $SO_2$ concentration in the atmospheric column derived from the ozone monitoring instrument (OMI) aboard Aura satellite for September 2014. High values near Iceland are due to the 2014–2015 flood lava eruption of the Bárðarbunga volcano. (Panel b) 2D normalized histogram of air parcel back trajectories arriving at the SMEAR II station for September 2014. The trajectories are computed using the HYSPLIT model going back 96 hours in time with a resolution of 9 arriving trajectories per hour. Each air parcel path is recorded for each hour. These points are binned in $2 \times 2$ degree cells. The counting of each cell is then normalized by multiplying it with the square of the distance to the SMEAR II station (black disk marker) in order to highlight the long-range transport patterns.

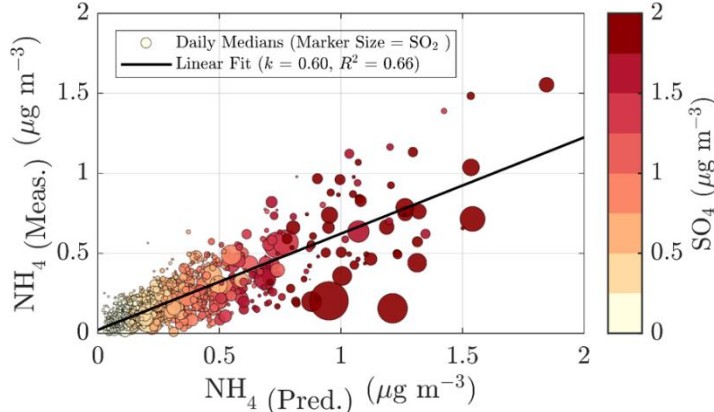

1255

**Figure A.7** The relationship between the measured and predicted ammonium concentrations. The marker size reflects the ambient
$SO_2$ concentration and the colour scale sulphate concentration. The linear fit represents the ratio between the measured and predicted
ammonium concentration. Drifting from 1 could be linked to more basic or acidic aerosols. The linear fit of 0.66 indicates a possibility
of acidic aerosols that decreases in the presence of $SO_2$. A better acidity approximation could be derived with thermodynamical models.

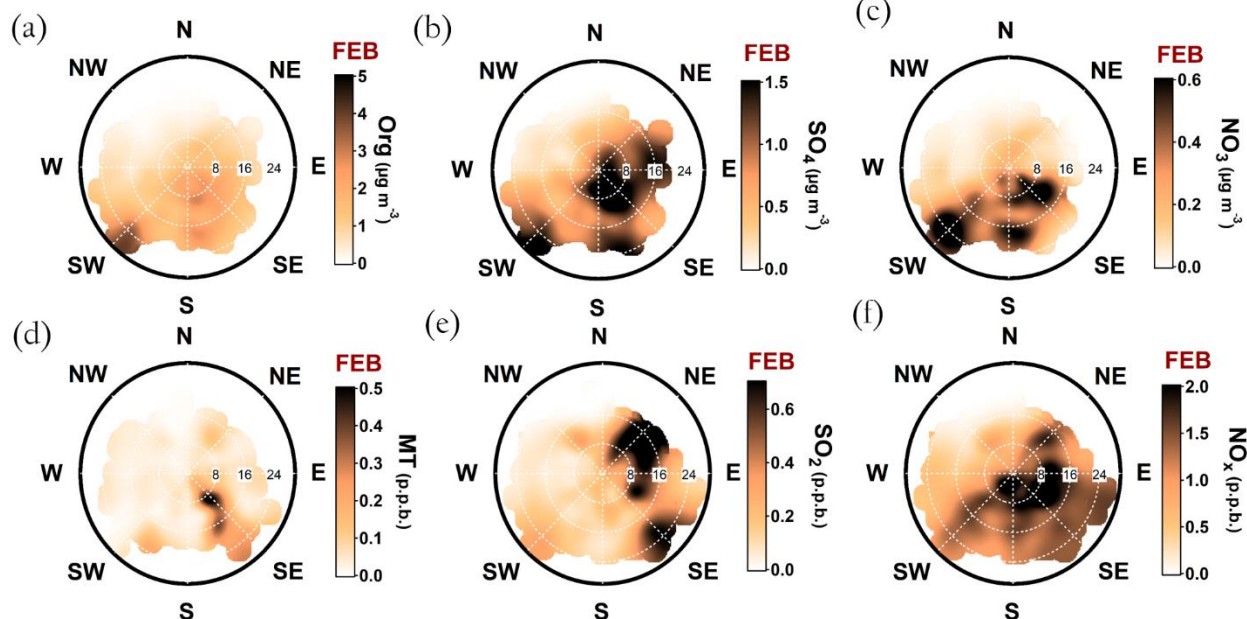

1260

**Figure A.8** Openair polar plots for organic aerosol (panel a), sulphate (panel b), nitrate (panel c), monoterpenes (panel d), SO₂ (panel e), and NOₓ (panel f) during February. The distances from the origin indicates wind speeds in km h⁻¹. The wind speed grid lines are presented with white dashed circles. The colour scales represent the concentrations observed with each wind speed and direction combinations. As the figures do not indicate any likelihood of the wind speed and distance combinations, Figure A.1 is important to keep in mind while interpreting them. Briefly, N–NE–E is the least probable wind direction, whereas S–SW–W is the most likely. Wind speeds generally stay below 20 km h⁻¹.

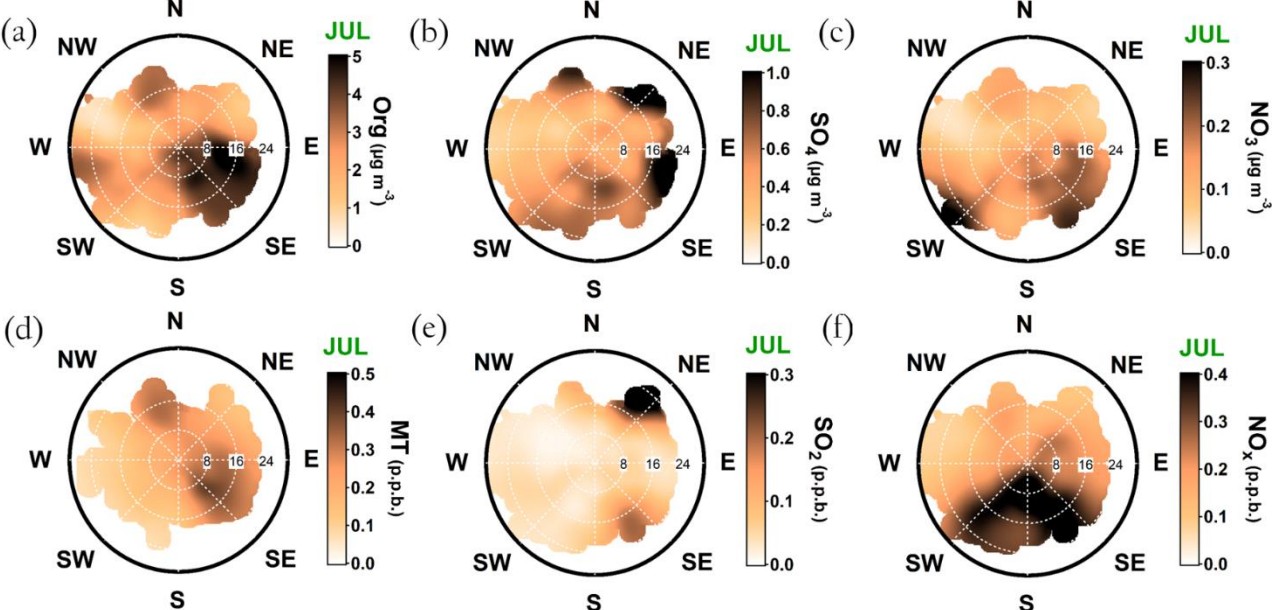

1267

**Figure A.9** Openair polar plots for organic aerosol (panel a), sulphate (panel b), nitrate (panel c), monoterpenes (panel d), SO₂ (panel e), and NOₓ (panel f) during July. The distances from the origin indicates wind speeds in km h⁻¹. The wind speed grid lines are presented with white dashed circles. The colour scales represent the concentrations observed with each wind speed and direction combinations. As the figures do not indicate any likelihood of the wind speed and distance combinations, Figure A.1 is important to keep in mind while interpreting them. Briefly, N–NE–E is the least probable wind direction, whereas S–SW–W is the most likely. Wind speeds generally stay below 20 km h⁻¹.