# Peer review of "Long-term sub-micron aerosol chemical composition in the boreal forest: inter- and intra-annual variability"

_Atmospheric Chemistry and Physics, 2019_

## Referee Comment (RC1) · Anonymous Referee #3 · 6 Dec 2019

The manuscript studies the sub-micron on-line aerosol composition at the research site of SMEAR II situated in the boreal forest of Finland for a long period spanning from 2012 to 2018 using an Aerosol Chemical Speciation Monitor to derive the inter- and intra-annual variability. Overall, organics represent the most abundant species, followed by sulphate, nitrate and ammonium. PM1 concentrations present a bimodal distribution peaking in February and in summer. The winter peak is mostly linked to enhanced inorganic components such as nitrate and sulphate, while the summer maximum is mostly linked to significant increase of organics, probably due to secondary organic aerosol formation. The study also takes into account parameters such as temperature, insolation, wind speed and direction when interpreting the diurnal and

seasonal patterns of the different aerosol components. Finally two case studies are examined, derived from this inter- and intra-annual variation, the enhanced concentrations during two summertimes (2014 and 2018) and enhanced sulphate loadings during September 2014.

The paper is well written and easy to follow, though there are some issues and more thorough discussion should be made in specific sections. Other than that the paper can be recommended for publication after addressing the issues listed below.

General comments:

- There is overall an inconsistency in figure numbering and their reference within the text e.g. P11L330 Figure 1a should be 2a and L338 should be figure 2b, P18L478 Figure4a should be 6a, P18L502 Figure d of which? Etc.

Specific comments:

- P12L361 It would be interesting to see whether OM/OC changes within the year as, e.g. SOA formation is expected to lead to more oxidized species and thus, higher OM/OC. I would suggest maybe color-coding Figure 2e based on the date. This would also be interesting further on in the manuscript, as 2018 is the case study having very warm summer (Section 3.1.1)

Technical corrections:

- P12L357-358 There seems to be something wrong with this sentence and what is inside the parenthesis.

- P18L504 only one week (delete "ca")

---

## Referee Comment (RC2) · Anonymous Referee #1 · 15 Dec 2019

This manuscript reports an ACSM measurement study of sub-micron particles conducted at the SMEAR II atmospheric research station of Finland for a period of 7 years from 2012 to 2018. Discussions are made on temporal, diurnal and seasonal variations of PM1 components, gaseous compounds including NOx, SO2, and monoterpene, and meteorological parameters such as temperature, solar radiation, wind speed and wind direction. Additionally, the influence of radiation, temperature and wind direction on major aerosol and gaseous species are examined. This is a worthy paper and a timely submission as it reports the longest online measurement data, to date, on sub-micron aerosol chemical composition in a boreal environment. It is suitable for publication on ACP and I recommend acceptance by the journal after the authors respond to the

following comments.

While the title of the paper highlights aerosol chemical composition, the discussions focus more heavily on the inter- and intra-annual variations of PM1 mass loading and meteorological conditions. The authors mention that more detailed discussions on organic aerosol factors determined from analysis of the ACSM mass spectra will be presented in a separate paper. While this decision is understandable considering the length of current manuscript, it is important that relevant discussions, such as biomass burning organic aerosols, are backed by measurement data such as variations in the ACSM f60 time series.

Another issue is that this manuscript cites a lot of previous publications from SMEAR II but sometime without providing sufficient contexts. Readers who are less familiar with the literature may feel somewhat lost or unconvinced. In addition, the creation of an Appendix section and the placement of several figures in there appear a bit haphazard and may introduce confusion. More detailed comments are given below:

What type of interpretation was applied to the image plots in Figure 1?

Line 182, chang "evaporating at 600oC maximum" to "flash evaporate at 600oC" .

Line 184, add "of" after "the signal".

Line 224, spell out the acronyms that have not yet been defined.

Line 307, it is briefly mentioned that large discrepancies between ACSM nitrate and MARGA nitrate were observed, likely introduced by organic nitrate. This is quite interesting, it would be helpful that the authors provide a bit more details and expand the discussions.

Line 316 – 317. Is this sentence referring to the CE values used in this study or those typically used for ACSM measurements?

Line 322, give details on how DMPS-derived mass concentration is determined

Lines 328, 330, change "Figure 1a" to "Figure 2a".

Check the texts at Line 357 – 358.

Figure 2 caption, check the text for (b)

Figure 4, what's the explanation for the large year-to-year varaitions in average SO2 concentration?

In 2017, SO2 was nearly 0 in all months but sulfate concentration was not too much different than those in the other years. Why so?

Line 476, "exceptionally long-lasting period with high atmospheric pressure", can this statement be a bit more quantitative, i.e., what does exceptionally long-lasting mean?

Line 479, quote the 7 year mean July temperature.

Line 699, what Figure c?

Line 702, revise this sentence " but shows . . ."

––––––––––––––––––––––––––––

---

## Author Comment (AC1) · 21 Jan 2020

**Author response to referee comments**

We thank the reviewers for their valuable inputs that helped to improve the manuscript. The manuscript was modified according to the reviewers' advices. The detailed line-by-line responses (written in black) to each referee report (reports are written in *blue*), respectively, as well as a list of the largest changes in the manuscript are listed below.

**Anonymous Referee #1**

*This manuscript reports an ACSM measurement study of sub-micron particles conducted at the SMEAR II atmospheric research station of Finland for a period of 7 years from 2012 to 2018. Discussions are made on temporal, diurnal and seasonal variations of $PM_1$ components, gaseous compounds including $NO_x$, $SO_2$, and monoterpene, and meteorological parameters such as temperature, solar radiation, wind speed and wind direction. Additionally, the influence of radiation, temperature and wind direction on major aerosol and gaseous species are examined. This is a worthy paper and a timely submission as it reports the longest online measurement data, to date, on sub-micron aerosol chemical composition in a boreal environment. It is suitable for publication on ACP and I recommend acceptance by the journal after the authors respond to the following comments.*

*While the title of the paper highlights aerosol chemical composition, the discussions focus more heavily on the inter- and intra-annual variations of $PM_1$ mass loading and meteorological conditions. The authors mention that more detailed discussions on organic aerosol factors determined from analysis of the ACSM mass spectra will be presented in a separate paper. While this decision is understandable considering the length of current manuscript, it is important that relevant discussions, such as biomass burning organic aerosols, are backed by measurement data such as variations in the ACSM $f_{60}$ time series.*

- The reviewer is right that $f_{60}$ serves as a good marker for biomass burning. It has been for long associated with levoglucosan-like species that result from cellulose pyrolysis (Schneider et al., 2006; Alfarra et al., 2007). The reviewer's statement is especially true in wintertime, when biomass burning organic aerosol (BBOA) is still fresh upon arrival to SMEAR II. The figures below (Figures AR.1&AR.3) clearly show how $f_{60}$ is making up a larger fraction of the organic aerosol in winter. As the summertime BBOA emissions occur mostly faraway (wild fires rarely occur in Finland), BBOA has already photochemically transformed into oxidised organic aerosol (OOA) before reaching SMEAR II, and $f_{60}$ has decreased to Northern hemispheric background levels ($f_{60} < \sim 0.03$; Cubison et al., 2011). Thus, $f_{60}$ is necessarily not a good marker for summertime OA origins, but rather for *fresh* BBOA. Hence, we originally chose eBC and CO as better (~inert) markers for summertime biomass burning influence. The rapid (in order of several hours) photochemical aging of BBOA has been a topic of earlier chamber (Grieshop et al., 2009; Jimenez et al., 2009; Cubison et al., 2011) and ambient studies (DeCarlo et al., 2010; Cubison et al., 2011).

  **Manuscript modifications:**

- We have added sub-panels to Figures 5&6 (Figures 5e&6e) in the manuscript to include $f_{60}$ data. The revised figures are presented below as Figure AR.1 and Figure AR.2. We also added information regarding the $f_{60}$ seasonal behaviour in form of Figure AR.3 in *Appendix A: Supporting figures* (Figure A.4 in the manuscript). Note the minor change in Figure 5 representation, as we wanted to include error bars (standard deviation) in the figure.

[Figure]

**Figure AR.1** The daily mean organic aerosol (Org, panel a), monoterpene (MT, panel b), carbon monoxide (CO, panel c), equivalent black carbon (eBC, panel d) concentrations, and the fraction of the ACSM Org-signal made up by levoglucosan-like species ($f_{60}$, panel e) recorded under different ambient temperatures. The values recorded assigned into 10 °C wide bins based on the daily ambient temperature mean. The marker error bars show the standard deviation of the values in each bin.

[Figure]

**Figure AR.2** Non–parametric probability densities (Kernel distributions) of temperature (panel a), organic aerosol (panel b), monoterpenes (panel c), equivalent black carbon (panel d) concentrations, and the fraction of the ACSM Org-signal made up by levoglucosan-like species ($f_{60}$; panel e) during individual Julys across the measurement period (2012–2018). The data availability for eBC was one week in July 2018. The x–axes represent the T, Org, MT and eBC values recorded, respectively and the y–axes the non–parametric probability densities. Briefly, the Kernel distributions are similar to smoothed histograms of the measurement data. This visualisation was chosen to avoid assumptions of the nature of distribution that might hide important features of the measurement data if presented with normal distributions, for instance.

[Figure]

**Figure AR.3** The mean (a-panel) and median (b-panel) $f_{60}$ (the fraction of $m/Q$ 60 Th signal of the total OA signal) values derived from the ACSM measurements (2012–2018 at SMEAR II). The x-axes represent the time of the year and the y-axes the hour of the day (UTC+2). The coloured pixels represent the $f_{60}$ values. Note the different colour scales between the mean and median figures. It is also worth mentioning that due to the rather low signal to noise ratio of the ACSM, the $f_{60}$ estimates can be very noisy. To avoid the weight of the high and low noise extremes in the a-panel (mean $f_{60}$), only the range of $0 \geq f_{60} \leq 1$ were included in the $f_{60}$ mean field calculation.

- The following text has been added to chapter *3.1 Inter- and intra-annual variation*:

**Page 17, lines 517–530:** *"While the quantification and separation of BBOA from SOA will be the topic of an upcoming independent publication centred on the analysis of organic aerosol mass spectral fingerprints at SMEAR II, we briefly introduce the behaviour of $f_{60}$. $f_{60}$, which equals the contribution of $m/Q$ 60 Th signal to the total organic signal recorded by the ACSM, is a marker for levoglucosan-like species originating from cellulose pyrolysis in biomass burning (Schneider et al., 2006;Alfarra et al., 2007). $f_{60}$ is present at high percentages in fresh BBOA plumes (Cubison et al., 2011), but decays due to BBOA photochemical aging into oxidised organic aerosol. The fairly rapid (in order of several hours to days) photochemical aging of BBOA has been a topic of earlier chamber (Grieshop et al., 2009;Jimenez et al., 2009) and ambient studies (DeCarlo et al., 2010;Cubison et al., 2011). Here, unlike CO and eBC, $f_{60}$ does not increase as a function of temperature in the highest temperature bins, but stays rather constant albeit lower than the $f_{60}$ values recorded under cold temperatures at SMEAR II (Figure 5e). As the possible wild fires contributing to SMEAR II CO and eBC under high ambient temperatures also occur further away, it is likely that the BBOA is oxidised before detected at SMEAR II. CO and eBC can be considered as more inert BBOA markers compared to $f_{60}$. The wintertime $f_{60}$ is likely linked to*

*wintertime biomass burning (for domestic heating purposes) emissions trapped in the shallow mixing layer. These emissions are discussed more later on in the manuscript (see chapter 3.2 Diurnal variation of NR–PM$_1$ composition)."*

- The following text has been added to chapter *3.1.1 The effect of warm summers on organic aerosol loading:*

**Pages 18–19, lines 579–589:** *"The f$_{60}$ in turn follows the conclusions made earlier in the context of Figure 5, as the f$_{60}$ values remain low each July, and approach the f$_{60}$ background levels of 0.3% (Cubison et al., 2011; Figure 6e). Importantly, such negligible f$_{60}$ signals were detected under the influence of an aged BBOA plume originating from Moscow and Northern Ukraine wild fires at SMEAR II also in 2010 (Corrigan et al., 2013). An AMS, which was used as one of the measurement tools in the campaign, detected mass spectra resembling oxidised organic aerosol during the biomass burning influence. These data correlated well with multiple biomass burning markers including CO, potassium and acetonitrile despite the lack of resemblance with fresh BBOA mass spectra with high f$_{60}$. Corrigan et al. (2013) finally attributed up to 25% of the organic aerosol to BBOA originating from the Moscow and Northern Ukraine wild fires. 35% of the organic aerosol mass was associated with biogenic SOA formation. The weather during the Corrigan et al. (2013) study period in 2010 was also unusually warm (T$_{avg}$ = 20°C), and resembled summers 2014 and 2018 also regarding the ruling anti-cyclonic circulation."*

- We also want to inform the reviewer that we have added a figure of the Org/OC-ratio (~OM/OC) monthly statistics from 2018 to the manuscript (Figure AR.6 which is Figure A.5 in the *Appendix A: Supporting figures*), as well as colour coded Figure 2e by month of the year (Figure AR.5 which is Figure 2e in the manuscript) due to the requests of Reviewer #3. The intra-annual variability of Org/OC also serves as additional chemical information regarding OA composition (~degree of oxygenation). These changes are specified below in the line-by-line response to Reviewer #3.

*Another issue is that this manuscript cites a lot of previous publications from SMEAR II but sometime without providing sufficient contexts. Readers who are less familiar with the literature may feel somewhat lost or unconvinced.*

- We agree with the reviewer.

**Manuscript modifications:**

- We have added the following text under chapter *1 Introduction:*

**Pages 4–5, lines 116–135:** *"The chemical composition of aerosol particles at SMEAR II has been studied previously in multiple short (<1 – 10 month) measurement campaigns with both offline (Saarikoski et al., 2005;Kourtchev et al., 2005;Cavalli et al., 2006;Finessi et al., 2012;Corrigan et al., 2013;Kourtchev et al., 2013;Kortelainen et al., 2017), and online methods (Allan et al., 2006;Finessi et al., 2012;Häkkinen et al., 2012;Corrigan et al., 2013;Crippa et al., 2014;Hong et al., 2014;Makkonen et al., 2014;Äijälä et al., 2017;Hong et al., 2017;Kortelainen et al., 2017;Riva et al., 2019;Äijälä et al., 2019). The previous studies include several important discoveries regarding SMEAR II aerosol composition. For example, a large mass fraction of particulate matter has been found to be (highly oxidised) organic aerosol (Corrigan et al., 2013;Crippa et al., 2014;Äijälä et al., 2017;Äijälä et al., 2019), and recognised as terpene oxidation products (Kourtchev et al., 2005;Allan et al., 2006;Cavalli et al., 2006;Finessi et al., 2012;Corrigan et al., 2013;Kourtchev et al., 2013). In addition to this forest-generated SOA, a nearby sawmill can also significantly contribute to the OA loading in the case of south easterly winds (e.g. Liao et al., 2011;Corrigan et al., 2013;Äijälä et al., 2017). The composition of the sawmill-OA is found to significantly resemble biogenic SOA (Äijälä et al., 2017). Also, biomass burning organic aerosol (BBOA) contributes to the OA mass (Corrigan et al., 2013; Crippa et al., 2014; Äijälä et al., 2017; Äijälä et al., 2019). The BBOA presence in the summertime aerosol depends on long-range transport of wildfire plumes. Inorganic species from anthropogenic activities, majority identified as ammonium sulphate and nitrate, are also transported to the station. Ammonium sulphate represents the dominating inorganic species (e.g. Saarikoski et al., 2005; Äijälä et al., 2019). Despite the long list of studies and discoveries, the measurement/sampling periods have occurred mostly between early spring and late autumn, leaving the*

*description of the wintertime aerosol composition nearly fully lacking. Hence, based on these studies alone, the understanding of the degree of variability in aerosol chemical composition at SMEAR II is incomplete – in both intra- and inter-annual scales."*

*In addition, the creation of an Appendix section and the placement of several figures in there appear a bit haphazard and may introduce confusion.*

- The purpose of the Appendix A is to provide supporting Figures to the main text.

**Manuscript modifications:**

- The Appendix A is now named appropriately: *Appendix A: Supporting figures.*

*More detailed comments are given below:*

*What type of interpretation was applied to the image plots in Figure 1?*

- The interpretation of Figure 1 is written in the section 2.1 under SMEAR II description. This section includes the description of the meteorological conditions ruling at SMEAR II.
- Just in case the reviewer meant to ask about the *interpolation* method applied to the image plots, we will clarify it, too: The figures were produced with the MATLAB 2017a *"contourf"* function. It performs the interpolation in a linear manner to create isolines of matrix **M**, where **M** contains values (temperature, global radiation, wind speed) on the *x-y* plane (month in year, hour in day). The selection of the isolines is performed automatically. Due to the circular nature of wind direction, no interpolation was applied there. Figure 1c was produced with MATLAB 2017a *"pcolor"* function. We now refer to the MATLAB 2017a *contourf-* and *pcolor-*functions in the Figure 1 caption.

**Manuscript modifications:**

- **Figure 1 caption (page 6, lines 185–191)**: *The seasonal evolution of diurnal cycles of ambient temperature measured 4.2 m above ground level (panel a), global radiation above the forest canopy (panel b), wind direction above the forest canopy (panel c), and wind speed above the forest canopy (panel d) recorded at SMEAR II station in 2012 – 2018. The y–axes in the figures represent the local time of day (UTC+2) and the x–axes the time of the year. The colour scales correspond to the temperature in degrees Celsius, global radiation in W m$^{-2}$, wind direction in degrees and wind speed in m s$^{-1}$, respectively. Panels a, b and d include interpolation of the 14 d × 1 h resolution data grid into isolines based on the MATLAB2017a contourf function. Panel c has no interpolation involved due to challenges related to interpolating over a circularly behaving variable. The plot is produced with MATLAB 2017a pcolor function.*

*Line 182, change "evaporating at 600°C maximum" to "flash evaporate at 600°C".*

**Manuscript modifications:**

- We changed the text as the reviewer suggested.

*Line 184, add "of" after "the signal".*

**Manuscript modifications:**

- We changed the text as the reviewer suggested.

*Line 224, spell out the acronyms that have not yet been defined.*

**Manuscript modifications:**

- We changed the text as the reviewer suggested.

*Line 307, it is briefly mentioned that large discrepancies between ACSM nitrate and MARGA nitrate were observed, likely introduced by organic nitrate. This is quite interesting, it would be helpful that the authors provide a bit more details and expand the discussions.*

- The differences between MARGA and AMS nitrate measurements at SMEAR II have been already discussed in Makkonen et al. (2014). They address i) noise introduced by low mass concentration of nitrate, ii) organic nitrates, and iii) the possibility of un-optimal MARGA nitrate background subtraction as possible factors leading to measurement discrepancies. We chose not to compare the MARGA and ACSM nitrate concentration under the chapter *2.5 ACSM chemical speciation validation* due to the known scatter between these measurements. Another possible factor weakening the nitrate correlation might also be a due to a time-to-time overestimation of the ACSM $NO_3$ caused by the presence of an organic fragment (i.e., $CH_2O^+$) coinciding with $NO^+$ at $m/Q$ 30 Th. Unfortunately, the unit mass resolution of the ACSM disables us from separating those ions, further forcing us to fully trust the default fragmentation table provided by Aerodyne Inc., the instrument manufacturer. However, as shown in Figure AR.3 low slope, this overestimation is likely not significant. Contrariwise, it is worth pointing out the differences in the ACSM and MARGA measurement size ranges (~$PM_1$ vs $PM_{2.5}$) that is the most likely reason for the low slope (k = 0.28) in the Figure AR.4 shown below.
- The presence of organic nitrates is discussed later in the manuscript in terms of the nitrate fragmentation ratio (*m/Q* 30 Th : *m/Q* 46 Th), and visualised in Figure 7g.

[Figure]

**Figure AR.4** The ACSM nitrate vs the $PM_{2.5}$ nitrate detected with MARGA–2S. The color coding represents the month of the year. The black line represents the overall linear fit. The figure can be found in the *Appendix A: Supporting figures*: Figure A.3.

**Manuscript modifications:**

- Figure AR.4, shown above, is added to *Appendix A: Supporting figures* (Figure A.3).
- The following text is added to chapter *2.5 ACSM chemical speciation validation*:

**Page 12, lines 398–409**: *"Regarding the water-soluble inorganic ions, only the $SO_4^{2-}$ concentration (in $PM_{2.5}$), retrieved from the MARGA measurements, was used for the current analysis for ACSM data validation purposes. The nitrate time series, for example, are known to be different between the two measurements at SMEAR II (Makkonen et al., 2014). The scatter between the nitrate measurements, visualised also here, in Figure A.3, could serve as evidence of organic nitrates, which are not efficiently detected by MARGA (Makkonen et al., 2014). The presence of organic nitrates is discussed later in the manuscript (see chapter 3.2 Diurnal variation of NR–$PM_1$ composition). Other factors influencing the nitrate agreement could arise from the MARGA nitrate background subtraction procedure, the overall low nitrate signal at SMEAR II (Makkonen et al., 2014), an organic $CH_2O^+$ fragment coinciding with $NO^+$ at m/Q 30 Th leading to ACSM nitrate over prediction under certain conditions, and finally the difference between the ACSM and MARGA size cuts ($PM_1$ vs $PM_{2.5}$). The ACSM sulphate, however, correlates well with MARGA (Pearson $R^2 = 0.77$), but has a slightly lower Pearson $R^2$ compared to an earlier < 11–month MARGA vs AMS comparison from SMEAR II (Pearson $R^2 = 0.91$) (Makkonen et al., 2014). Overall, based on the good agreement between ACSM and Sunset OCEC, MARGA, DMPS and Dekati cascade impactor measurements, we are confident of the year–to–year comparability of our ACSM dataset."*

- The following text is added to chapter *3.2 Diurnal variation of NR-$PM_1$ composition*:

**Page 21, lines 676–678**: *"Such high organic nitrate fraction could also explain some of the scatter observed in the ACSM $NO_3$ and MARGA $NO_3$ comparison discussed earlier in the manuscript (Figure A.3 and chapter 2.5 ACSM chemical speciation validation)."*

*Line 316 – 317. Is this sentence referring to the CE values used in this study or those typically used for ACSM measurements?*

- The CE values refer to those typically used for AMS type of instrument.

**Manuscript modifications:**

- We clarified the statement.

*Line 322, give details on how DMPS-derived mass concentration is determined*

**Manuscript modifications:**

- The following details are now given in chapter *2.4 ACSM collection efficiency correction:*

  **Page 20, lines 351–356:** *"The DMPS-derived mass concentration is determined as follows: i) calculation of aerosol volume concentration ($m^3$ $m^{-3}$) of the ACSM detectable size range, where the aerodynamics lens transmission is most efficient (50–450 nm in electrical mobility = ~75–650 nm in vacuum aerodynamic diameter) and assuming spherical particles, ii) estimating aerosol density based on the ACSM-measured chemical composition ($\rho_{(NH_4)_2SO_4}$= 1.77 g $cm^{-3}$, $\rho_{NH_4NO_3}$= 1.72 g $cm^{-3}$, $\rho_{Org}$= 1.50 g $cm^{-3}$, $\rho_{BC}$ = 1.00 g $cm^{-3}$), iii) calculating the mass concentration (μg $m^{-3}$; mass concentration = density × volume concentration)."*

*Lines 328, 330, change "Figure 1a" to "Figure 2a".*

**Manuscript modifications:**

- All the figure numberings were checked and corrected.

*Check the texts at Line 357 – 358.*

**Manuscript modifications:**

- The misplaced text was deleted.

*Figure 2 caption, check the text for (b)*

**Manuscript modifications:**

- The misplaced text was deleted.

*Figure 4, what's the explanation for the large year-to-year variations in average $SO_2$ concentration?*

- The $SO_2$ concentration at SMEAR II is controlled by $SO_2$ sources (emissions), sinks (photochemistry/heterogeneous oxidation to sulphate, wet and dry deposition), transport (general circulation affecting wind direction and speed as well as $SO_2$ plume dispersion) and boundary layer meteorology (temperature inversions trapping pollutants). The intra-annual variability of $SO_2$ concentration at SMEAR II is likely a result from the variability of one or many of these factors. As can be seen from Figure 9e in the manuscript, the elevated $SO_2$ concentrations (dark areas in the concentration field) are associated with very specific, rather narrow ranges of ~easterly wind directions, and elevated wind speeds. This figure alone visualises that the SMEAR II $SO_2$ concentration is *very sensitive* to even moderate wind direction (~analogous with air mass trajectory) and speed variations. The Figure AR.5 demonstrates the wind direction variability in different Februaries throughout the measurement period. The yellow shadings indicate the approx. wind direction areas associated with elevated $SO_2$ loadings. The year-to-year variability in wind direction is significant, which can certainly explain some of the intra-annual variability visualised in Figure 4 just in February. A more detailed answer would require a comprehensive investigation on the variability of the different factors affecting $SO_2$ concentration that would require emission inventories, and modelling efforts.

[Figure]

**Figure AR.5** Wind direction histograms for each February of the measurement period (2012—2018). The yellow shaded areas indicate wind directions associated with high $SO_2$ anomalies at SMEAR II (Figure 9e). Note that this figure does not visualize the wind speed data that is often needed to exceed 20 km h$^{-1}$ for the $SO_2$ transport to occur efficiently (Figure 9e).

**Manuscript modifications:**

- We have added the following text to chapter *3.3.3 Openair: Sulphate and SO₂*:

**Page 27, lines 796–802**: *"As can be seen from Figures 9e, A.8e & A.9e, elevated $SO_2$ concentrations (dark areas in the concentration fields) are associated with very specific, rather narrow ranges of easterly (mainly NE and SE) wind directions, and elevated wind speeds (> 16 – 20 km h$^{-1}$). These figures illustrate the sensitivity of the recorded $SO_2$ concentration towards even moderate wind direction and speed variations. As wind direction and speed can vary significantly in an inter-annual scale, also inter-annual variability in $SO_2$ concentration can be expected. This could finally explain the $SO_2$ inter-annual variability highlighted especially in winter months (Figure 4e). Such strong variability is not reflected further in the sulphate aerosol (Figure 4b) year-to-year scales due to its long lifetime and build-up in the atmosphere."*

*In 2017, SO₂ was nearly 0 in all months but sulfate concentration was not too much different than those in the other years. Why so?*

- The link between $SO_2$ and sulfate is not easily shown with ambient data due to the long lifetime/ atmospheric build-up of sulfate aerosol, and the sensitivity of $SO_2$ concentration towards meteorological variability. This results in quite poor correlation between $SO_2$ and $SO_4$-aerosol at SMEAR II ($R^2 = 0.14$; Figure AR.6). The $SO_2$ concentration is generally elevated with easterly winds (Figure 9e in the manuscript), but also requires relatively high wind speeds (>20 km h$^{-1}$; Figure 9e) to diminish the $SO_2$ concentration decline during transport via photochemical or wet deposition sink pathways. The discrepancy mentioned by the reviewer between the detected $SO_2$ and $SO_4$ concentrations might be quantitatively explained with a comprehensive $SO_2$ emission inventory map together with detailed investigation of the meteorological conditions.

[Figure]

**Figure AR.6** The relationship between the $SO_2$ and particulate $SO_4$ concentrations at SMEAR II. Note the logarithmic scale on the x-axis.

**Manuscript modifications:**

- The topic has been reflected in the previous manuscript modification made:

**Page 27, lines 799–802:** *"As wind direction and speed can vary significantly in an inter-annual scale, also inter-annual variability in $SO_2$ concentration can be expected. This could finally explain the $SO_2$ inter-annual variability highlighted especially in winter months (Figure 4e). Such strong variability is not reflected further in the sulphate aerosol (Figure 4b) year-to-year scales due to its long lifetime and build-up in the atmosphere."*

*Line 476, "exceptionally long-lasting period with high atmospheric pressure", can this statement be a bit more quantitative, i.e., what does exceptionally long-lasting mean?*

- A more quantitative statement would require more detailed meteorological analysis on the pressure anomalies.

**Manuscript modifications:**

- We simplified the sentence and added a reference:

**Page 17, lines 540–542**: *"These summers were the hottest during the whole measurement period (**Error! Reference source not found.**6a), and linked to persistent high pressure conditions (Sinclair et al., 2019;FMI, 2014) "*.

*Line 479, quote the 7 year mean July temperature.*

**Manuscript modifications:**

- Quoted.

*Line 699, what Figure c?*

**Manuscript modifications:**

- All the figure numberings were checked and corrected.

*Line 702, revise this sentence " but shows . . ."*

**Manuscript modifications:**

- Revised.

**Anonymous Referee #3**

*The manuscript studies the sub-micron on-line aerosol composition at the research site of SMEAR II situated in the boreal forest of Finland for a long period spanning from 2012 to 2018 using an Aerosol Chemical Speciation Monitor to derive the inter- and intra-annual variability. Overall, organics represent the most abundant species, followed by sulphate, nitrate and ammonium. PM₁ concentrations present a bimodal distribution peaking in February and in summer. The winter peak is mostly linked to enhanced inorganic components such as nitrate and sulphate, while the summer maximum is mostly linked to significant increase of organics, probably due to secondary organic aerosol formation. The study also takes into account parameters such as temperature, insolation, wind speed and direction when interpreting the diurnal and seasonal patterns of the different aerosol components. Finally two case studies are examined, derived from this inter- and intra-annual variation, the enhanced concentrations during two summertimes (2014 and 2018) and enhanced sulphate loadings during September 2014.*

*The paper is well written and easy to follow, though there are some issues and more thorough discussion should be made in specific sections. Other than that the paper can be recommended for publication after addressing the issues listed below.*

*General comments:*

*- There is overall an inconsistency in figure numbering and their reference within the text e.g. P11L330 Figure 1a should be 2a and L338 should be figure 2b, P18L478 Figure4a should be 6a, P18L502 Figure d of which? Etc.*

**Manuscript modifications:**

- All of the figure numberings checked and corrected.

*Specific comments:*

*- P12L361 It would be interesting to see whether OM/OC changes within the year as, e.g. SOA formation is expected to lead to more oxidized species and thus, higher OM/OC. I would suggest maybe color-coding Figure 2e based on the date. This would also be interesting further on in the manuscript, as 2018 is the case study having very warm summer (Section 3.1.1)*

- The manuscript Figure 2e is now updated according to the reviewer's wishes (Figure AR.7). It now includes color-coding based on the month of the year. Highest slopes ($k$) corresponding to the OM:OC are observed in summer ($k = 1.65$ in July). The variability of slopes is rather small, as also wintertime OA is mainly highly aged aerosol, notably also in winter (low-volatility oxygenated organic aerosol, LV-OOA; Heikkinen et al., 2020, *in prep*).

[Figure]

**Figure AR.7** Updated manuscript Figure 2e. Organic carbon concentration (OC) vs the organic aerosol concentration (Org) in 2018 at SMEAR II.

- As the OCEC Analyser was not functioning optimally most of the time, we are not including other years to the analysis. Otherwise we could have compared the slopes (Org:OC values) in different summers to see if some summers could have more influence form biomass burning, for example. However, if we plot the Org:OC mean/median values, we observe a higher value in August 2018 compared to July 2018 which could indicate a higher contribution of less oxidised OA in July (Figure AR.8).

[Figure]

**Figure AR.8** Organic carbon concentration (OC) vs the organic aerosol concentration (Org) in 2018 at SMEAR II. The darker yellow shadings indicate the area between the 25th and 75th percentiles and the lighter yellow the area between the 10th and 90th percentiles. The figure is added under *Appendix A: Supporting figures* as Figure A.5.

**Manuscript modifications:**

- Figure 2e updated with Figure AR.7 shown above.
- Figure AR.8 added to *Appendix A: Supporting figures* as Figure A.5.
- The following text is added to chapter *3.1.1 Case study: The effect of warms summers on organic aerosol loading*:

    **Page 18, lines 567–572:** *"A quick revisit to the OC/EC vs ACSM comparison performed earlier in the manuscript (Figure 2e) shows relatively high Org/OC-values (k = 1.68) for July 2018, which further indicates of high oxygenation of OA (Aiken et al., 2008). However, the time series of the ruling monthly Org/OC-values visualised in Figure A.5 reveals an even higher Org/OC for August 2018. Further analysis of Org/OC recorded from SMEAR II is needed to answer whether such behaviour is frequently occurring at SMEAR II, or whether July 2018 organic aerosol was less functionalised (oxidised) than usual due to higher presence of primary organic aerosol, such as BBOA, than usual."*

- The following text is added to chapter *3.2 Diurnal variation in NR-PM$_1$ composition*:

    **Page 20, lines 654–656:** *"Such conclusions can also be made based on the rather high Org/OC linear regression slope for February data (k = 1.65) depicted in Figure 2e (see also Figure A.5). As mentioned earlier, a high Org/OC-ratio indicates a higher degree of functionalisation/oxidation of organic aerosol (Aiken et al., 2008)."*

*Technical corrections:*

*- P12L357-358 There seems to be something wrong with this sentence and what is inside the parenthesis.*

**Manuscript modifications:**

- Misplaced text removed.

*- P18L504 only one week (delete "ca")*

**Manuscript modifications:**

- 'ca' removed from the sentence.

**Other manuscript modifications**

-   The following text was added to chapter *3.3.3 Openair: Sulphate and SO₂* to include few Finnish national SO₂ sources to the discussion:

[revised manuscript text omitted]